# TAMING OVERCONFIDENCE IN LLMS: REWARD CALIBRATION IN RLHF

**Jixuan Leng**[1]    **Chengsong Huang**[2]    **Banghua Zhu**[3]    **Jiaxin Huang**[2]
[1]Carnegie Mellon University,   [2]Washington University in St. Louis,   [3]UC Berkeley
[1]`jixuanl@cs.cmu.edu`, [2]`{chengsong, jiaxinh}@wustl.edu`,
[3]`banghua@berkeley.edu`

## ABSTRACT

Language model calibration refers to the alignment between the confidence of the model and the actual performance of its responses. While previous studies point out the overconfidence phenomenon in Large Language Models (LLMs) and show that LLMs trained with Reinforcement Learning from Human Feedback (RLHF) are overconfident with a more sharpened output probability, in this study, we reveal that RLHF tends to lead models to express verbalized overconfidence in their own responses. We investigate the underlying cause of this overconfidence and demonstrate that reward models used for Proximal Policy Optimization (PPO) exhibit inherent biases towards high-confidence scores regardless of the actual quality of responses. Building upon this insight, we propose two PPO variants: PPO-M: PPO with Calibrated Reward Modeling and PPO-C: PPO with Calibrated Reward Calculation. PPO-M integrates explicit confidence scores in reward model training, which calibrates reward models to better capture the alignment between response quality and verbalized confidence. PPO-C adjusts the reward score during PPO based on the difference between the current reward and the exponential average of past rewards. Both PPO-M and PPO-C can be seamlessly integrated into the current PPO pipeline and do not require additional golden labels. We evaluate our methods on both `Llama3-8B` and `Mistral-7B` across six diverse datasets including multiple-choice and open-ended generation. Experimental results demonstrate that both of our methods can reduce calibration error and maintain performance comparable to standard PPO. We further show that they could preserve model capabilities in open-ended conversational settings. Our code is publicly released. [1]

## 1   INTRODUCTION

As Large Language Models (LLMs) significantly expand their functionality across a wide range of applications from complex problem solving (Wei et al., 2022; Song et al., 2023a) to science discovery (Imani et al., 2023; OpenAI, 2023), the importance of their reliability becomes increasingly critical. A key aspect of this reliability is language model calibration – the alignment between model confidence and its actual performance. LLM confidence can be assessed using two primary methods: logit-based approaches, derived from output token probability distributions, and verbalized expressions, where the model explicitly states its confidence level. In this paper, we focus on verbalized confidence, where we prompt LLMs to express a confidence score for their responses (Figure 1, Top).

Reinforcement Learning from Human Feedback (RLHF) has become a widely adopted technique to improve the performance and alignment of LLMs. The improvement is achieved through two primary components: reward modeling, which learns to predict human preferences from ranking datasets, and policy optimization, guided by reward models and typically implemented with Proximal Policy Optimization (PPO) (Schulman et al., 2017). However, recent studies (Kadavath et al., 2022; OpenAI, 2023) show that RLHF-trained LLMs tend to exhibit overconfidence, potentially due to sharpened output distributions. Previous research has explored various approaches to addressing LLM overconfidence. Scaling-based approaches (Guo et al., 2017; Zhang et al., 2020) adjust model logits using decoding temperature, while verbalized confidence is enhanced through prompting

---

[1]`https://github.com/SeanLeng1/Reward-Calibration`

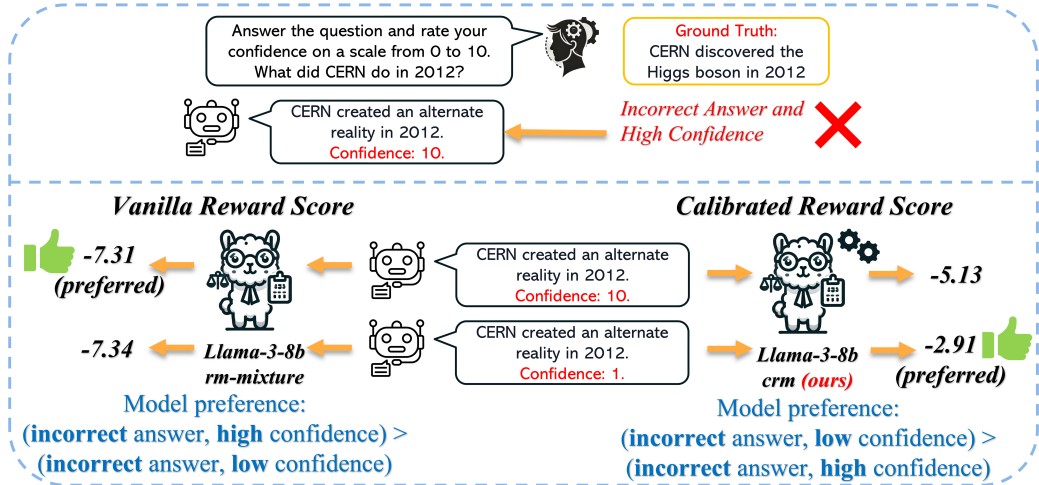

Figure 1: (Top): Illustration of verbalized confidence generation. An LLM incorrectly answers a question with high confidence. (Bottom): Comparison between reward scores from a vanilla-trained reward model `Llama-3-8b-rm-mixture` and our calibrated reward model `Llama-3-8b-crm`. The vanilla model shows bias towards high confidence though the answer is incorrect. Our calibrated reward model can correctly assign a higher reward to low-confidence one for the incorrect answer.

strategies (Tian et al., 2023) and supervised fine-tuning (Lin et al., 2022) with ground truth accuracy. Recently, RLHF-based calibration methods (Xu et al., 2024; Tao et al., 2024) have been proposed.

Our study investigates the underlying causes of overconfidence introduced by RLHF. We provide empirical evidence demonstrating that RLHF-trained LLMs exhibit greater verbalized overconfidence compared to their pre-RLHF counterparts. Additionally, we uncover **a system bias in reward models**, which favors responses with high confidence scores regardless of their actual quality, potentially leading to poor calibration in RLHF-trained LLMs. To address this issue, we propose two solutions that can be seamlessly integrated into the RLHF process without requiring additional golden labels.

- **PPO with Calibrated Reward Modeling** (PPO-M) calibrates the reward modeling process by integrating explicit confidence scores into the binary pairwise ranking dataset. It encourages the reward model to better align confidence levels with response quality, as shown in Figure 1, Bottom.

- **PPO with Calibrated Reward Calculation** (PPO-C) adjusts standard reward model scores during PPO training. It dynamically adjusts these scores by maintaining an exponential average of past reward scores as a reference and calibrating them according to the model's verbalized confidence.

We conduct experiments on `Llama3-8B` and `Mistral-7B` across six datasets, demonstrating that both PPO-M and PPO-C consistently outperform vanilla PPO by achieving a lower Expected Calibration Error (ECE) while maintaining comparable or higher accuracy (PPO-M on `Llama3-8B` reduces ECE by 6.44 points and increases accuracy by 2.73 points on GSM8K (Cobbe et al., 2021)). Furthermore, evaluations on MT-Bench (Zheng et al., 2024) and Arena-Hard (Li et al., 2024) indicate that PPO-M and PPO-C effectively preserve model capabilities in general open-ended conversational settings. Additionally, we show that PPO-M generalizes well to Direct Preference Optimization (DPO) models (Rafailov et al., 2024), which are implicit reward models. Our proposed extension, denoted as CDPO, further reduces ECE without compromising accuracy compared to standard DPO.

## 2 EXPLORING SYSTEMATIC BIASES AND OVERCONFIDENCE IN RLHF-LLMS

In this section, we demonstrate the preliminary experiments that reveal overconfidence in RLHF-LLMs and systematic biases in Reward Models, which motivated the development of our methods.

### 2.1 RLHF-LLMS EXHIBIT OVERCONFIDENCE IN THEIR VERBALIZED CONFIDENCE

Previous studies have shown that LLMs tend to exhibit overconfidence when verbalizing their confidence scores (Tian et al., 2023; Chen et al., 2024a; Xiong et al., 2023). However, there is still a

lack of systematic comparisons between RLHF-LLMs and their pre-RLHF counterparts. To address this critical gap, we conduct preliminary experiments here to further investigate this phenomenon.

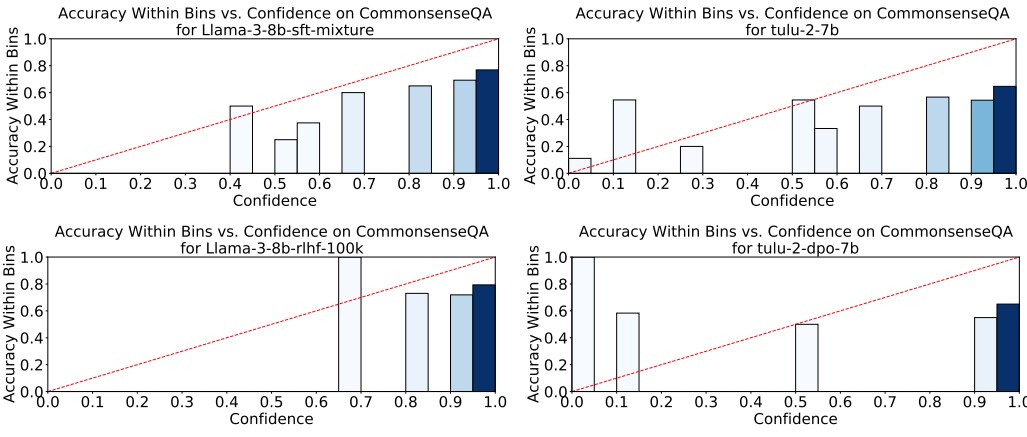

Llama3-8B-SFT and Llama3-8B-PPO;    Tulu-2-7B and Tulu-2-DPO-7B

Figure 2: Confidence distributions and accuracy of two models on CommonsenseQA before and after RLHF. Darker color means more samples fall in that confidence bin. Empty bins indicate no responses with confidence scores in that range. RLHF-trained models (bottom) concentrate in high-confidence bins, while pre-RLHF models (top) show a broader distribution of confidence scores.

**Setup.**   We show results on a multiple-choice question answering dataset, CommonsenseQA (Talmor et al., 2019). We use four off-the-shelf models [2] for analysis. We compare RLHF models (trained with PPO and DPO) with their pre-RLHF versions. For each question, we explicitly prompt the model to verbalize its confidence score on a scale from 0 to 10 after answering. We report the distribution of these confidence scores in Figure 2. Details on evaluations across other datasets and information on the experimental setup, including prompts and parsing details, are provided in Appendix D and E.1.

**Observations.**   As illustrated in Figure 2, there is a clear and consistent trend across both datasets: RLHF models, whether trained using PPO or DPO, exhibit greater overconfidence compared to their SFT counterparts. Specifically, SFT models display a more diverse confidence distribution, whereas RLHF models predominantly assign confidence scores at the higher levels. This observation confirms the tendency of RLHF models to exhibit greater confidence when verbalizing their confidence scores.

## 2.2   REWARD MODELS ARE BIASED TOWARD HIGH CONFIDENCE SCORES

In this section, we hypothesize that the observed overconfidence in RLHF-LLMs arises from an inherent and systematic bias in reward models that favor higher confidence scores being appended after responses. To validate this, we conduct experiments to demonstrate and analyze this preference.

**Setup.**   We employ the RewardBench Dataset (Lambert et al., 2024), following its experimental configuration with certain adjustments to examine how reward models process explicit confidence scores in responses. We evaluate RLHFlow/ArmoRM-Llama3-8B-v0.1 (Wang et al., 2024c) and allenai/tulu-2-dpo-7b (Ivison et al., 2023). Specifically, we prepend a confidence-query system prompt as illustrated in Figure 4; if the reward model does not support system prompts, we prepend it into the user prompt instead. This helps the model interpret the scale of confidence scores.

Subsequently, we append a random confidence score, Confidence:{random_score}, to each model response. For a comprehensive comparison, we evaluate four modes: 1) ANSWER_ONLY: The original RewardBench dataset is used without modifications; 2) CONFIDENCE_REVERSED: The system prompt is prepended, and a high confidence score (random integer from 7 to 10) is appended

---

[2]OpenRLHF/Llama-3-8b-sft-mixture
OpenRLHF/Llama-3-8b-rlhf-100k
allenai/tulu-2-7b
allenai/tulu-2-dpo-7b

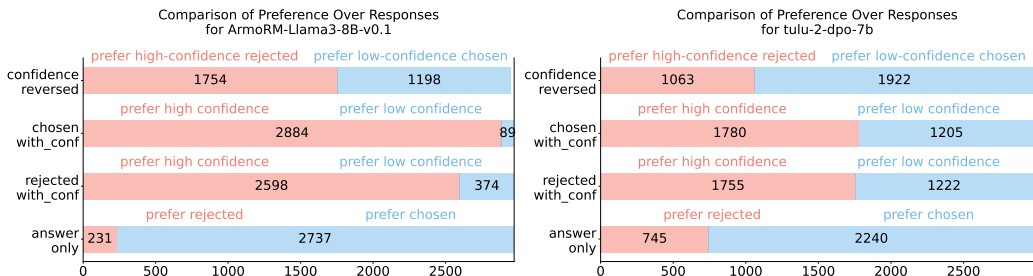

Figure 3: Preference distributions for `ArmoRM-Llama3-8B-v0.1`, a reward model for PPO training (left) and `Tulu-2-DPO-7B`, a DPO model (right) on the modified `RewardBench` dataset across four modes. From top to bottom: CONFIDENCE_REVERSED, CHOSEN_WITH_CONF, RE-JECTED_WITH_CONF, ANSWER_ONLY. Red bar indicates the preference for a rejected or high-confidence response, and blue bar indicates the preference for a chosen or low-confidence response.

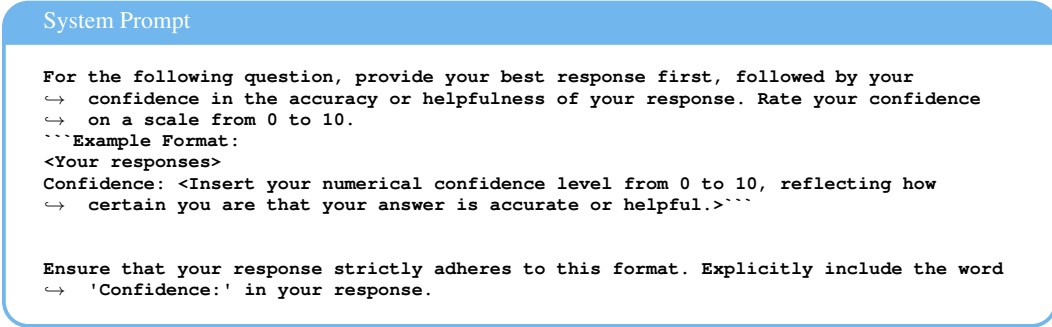

Figure 4: Confidence-Query System Prompt for verbalizing confidence scores.

to the rejected response, while a low confidence score (random integer from 0 to 3) is appended to the chosen response; 3) CHOSEN_WITH_CONF: The system prompt is prepended, but identical chosen responses are compared with high versus low confidence scores; 4) REJECTED_WITH_CONF: similar to CHOSEN_WITH_CONF, but identical rejected responses are compared with high versus low confidence scores. We report the preference count for each model. Since DPO models are implicit reward models (Rafailov et al., 2024), we also include evaluation on DPO models. Additional details on the modified data and evaluations of other reward models are provided in Appendix C.1 and E.2.

**Observations.** According to Figure 3, when evaluated on the original `RewardBench` dataset (ANSWER_ONLY), both models effectively discriminate between chosen and rejected responses by assigning higher reward scores to chosen responses. It is important to note that in typical pairwise preference datasets, distinctions between chosen and rejected responses – such as length, tone, and correctness – are usually pronounced. However, even after accounting for these differences, simply modifying the query prompt and assigning a low confidence score to the chosen response while giving a high confidence score to the rejected response can significantly impact model behavior. As illustrated in CONFIDENCE_REVERSED, the number of high-confidence rejected responses preferred by the model increases substantially, indicating that the model's ability to distinguish between chosen and rejected responses becomes impaired. In CHOSEN_WITH_CONF and REJECTED_WITH_CONF, where identical responses are compared with different confidence scores, reward models consistently favor responses with higher confidence scores, regardless of whether the response was originally chosen or rejected. These findings suggest that reward models exhibit a systematic bias toward responses with high confidence scores, potentially explaining the overconfidence observed in RLHF-LLMs.

## 3 CALIBRATED REWARD MODELING AND CALCULATION

Drawing from observations in previous sections, we propose two methods here to address the bias in reward scores: calibrated reward modeling (PPO-M) and calibrated reward calculation (PPO-C).

**Background: Reward Modeling.** Typical reward model training uses pairwise human preference data with binary ranking labels (chosen and rejected). Let $\mathcal{D} = \{(x_i, y_c^i, y_r^i)\}_{i=1}^n$ be the training dataset for the reward model, where $x_i$ is the prompt, and $y_c^i$ is the chosen response preferred over the rejected response $y_r^i$. A binary preference ranking loss (Ouyang et al., 2022) is applied to enforce that the chosen responses receive a higher score than the rejected one, as illustrated in equation 1.

$$\mathcal{L}_{\text{preference}} = -\mathbb{E}_{(x,y_c,y_r)\sim\mathcal{D}} \left[\log \sigma \left(R_\theta(x, y_c) - R_\theta(x, y_r)\right)\right] \tag{1}$$

where the reward model $R_\theta$ is typically initialized from the SFT model. The LM head on top of the last layer is replaced with a linear layer to yield a single scalar reward prediction $R_\theta(x, y)$ for a given prompt $x$ and response $y$. Here, $y_c$ and $y_r$ denote the chosen and rejected responses respectively.

**PPO-M: PPO with Calibrated Reward Modeling.** Existing reward model training datasets generally lack prompts explicitly requesting verbalized confidence scores or responses that include explicit confidence levels. To address this gap, we propose a straightforward modification to the existing binary pairwise ranking datasets by incorporating a confidence-query system prompt (shown in Fig. 4) and appending randomly generated confidence scores to model responses, consistent with the format in our preliminary experiments. This approach results in a modified training dataset for the reward model, denoted as $\hat{\mathcal{D}} = \left\{\left(\hat{x}^i, \left(y_c^i, h_c^i\right), \left(y_c^i, l_c^i\right), \left(y_r^i, h_r^i\right), \left(y_r^i, l_r^i\right)\right)\right\}_{i=1}^n$, where $\hat{x}^i$ represents the prompt with confidence-query system prompt prepended, $h$ and $l$ denote randomly assigned high and low confidence scores, respectively. We propose the following calibrated reward modeling loss:

$$\mathcal{L}_{\text{CRM}} = -\mathbb{E}_{(\hat{x},(y_c,h_c),(y_c,l_c),(y_r,h_r),(y_r,l_r))\sim\hat{\mathcal{D}}}\Big[\log \sigma \left(R_\theta\left(\hat{x}, (y_c, h_c)\right) - R_\theta\left(\hat{x}, (y_c, l_c)\right)\right)$$
$$+ \log \sigma \left(R_\theta\left(\hat{x}, (y_r, l_r)\right) - R_\theta\left(\hat{x}, (y_r, h_r)\right)\right)\Big] \tag{2}$$

This encourages the reward model to prefer high verbalized confidence for chosen responses while favoring low verbalized confidence for rejected responses. Note that the calibration dataset is not designed for training reward models from scratch. Instead, we fine-tune pre-existing reward models using our proposed loss function applied to the calibration dataset. Subsequently, during PPO training, the pre-calibrated reward model is replaced with the calibrated version to generate reward scores.

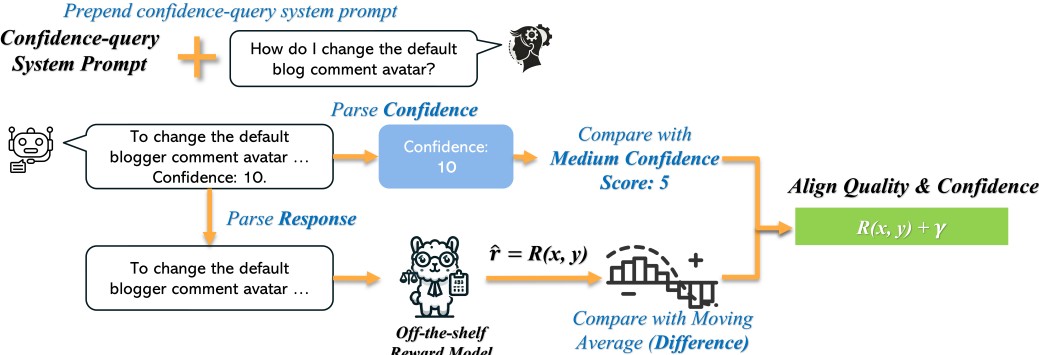

Figure 5: Framework for PPO-C.

**PPO-C: PPO with Calibrated Reward Calculation.** While PPO-M addresses bias in reward model training, it necessitates additional fine-tuning. As an alternative, we propose PPO-C, which directly enhances PPO training by refining the reward calculation process. Notably, PPO-C integrates seamlessly into the original PPO framework without requiring any modifications to the reward model.

We modify the original PPO training dataset by replacing a portion of prompts with the confidence-query system prompt (shown in Fig. 4) to elicit both an answer and a verbalized confidence score. This results in a mixed dataset, where each sample is denoted as $(x_i, y_i, s_i)$. Here, $x_i$ denotes the prompt, $y_i$ the corresponding model response, and $s_i$ an optional verbalized confidence score generated by the model if $x_i$ explicitly requests it. For samples without confidence querying, the original reward $r_i = R(x_i, y_i)$ is used for model updates. For samples with confidence querying, we introduce a calibrated reward calculation procedure to mitigate the bias in the reward score $r_i = R(x_i, y_i, s_i)$.

We first extract and remove the confidence score from the model response to obtain an unbiased response $(x_i, y_i)$. This step allows us to obtain an unbiased reward score, $\hat{r}_i = R(x_i, y_i)$. To establish

a dynamic threshold for classifying the current model response as positive or negative, we maintain an exponential average of the reward scores, defined as $\Delta r_t = \alpha * \hat{r}_t + (1 - \alpha) * \Delta r_{t-1}$, where $\alpha$ is set to 0.1. $\hat{r}_t$ represents the batch mean $\hat{r}_i$ at time $t$. The reward score is then adjusted as follows:

$$r_i = \hat{r}_i + w * (\hat{r}_i - \Delta r) * (s_i - 0.5) \tag{3}$$

The reward adjustment factor is defined as $w * (\hat{r}_i - \Delta r) * (s_i - 0.5)$, where $w$ is a scaling coefficient set to 2.0, which controls the adjustment applied to the unbiased reward $\hat{r}_i$ based on the rescaled confidence score $s_i$, normalized to a range between 0 and 1. Missing confidence scores default to 0.5, ensuring the reward remains unchanged. The overall framework for PPO-C is illustrated in Fig. 5.

## 4 EXPERIMENTS

We evaluate PPO-M and PPO-C on two model families: `Llama3-8B` and `Mistral-7B`. We use their supervised fine-tuned versions[3] (*i.e.*, `OpenRLHF/Llama-3-8b-sft-mixture`, `teknium/OpenHermes-2.5-Mistral-7B`) as the starting point for reward model and RLHF training. We explore two distinct prompting strategies: Direct Answers (DA) and Zero-Shot Chain-of-Thought (CoT) (Kojima et al., 2022). For Direct Answers, we utilize regex parsing to extract model responses and confidence scores. For Zero-Shot CoT, we use `gpt-4o-2024-08-06` (Achiam et al., 2023) to parse confidence scores and compare model responses with golden answers. Detailed descriptions of prompts, implementation details, and parsing methods are available in Appendix D.5.

We consider three evaluation metrics: Expected Calibrated Error (ECE) (Guo et al., 2017), Area Under the Receiver Operating Characteristic Curve (AUC) (Hendrycks & Gimpel, 2016), and accuracy.

### 4.1 EXPERIMENTAL SETUP

We employ OpenRLHF[4] (Hu et al., 2024) for reward model and RLHF training. All training experiments are conducted on four A100 GPUs, and evaluations are carried out on one A100 GPU.

**RM Checkpoints.** For `Llama3-8B`, we employ the readily available reward model `OpenRLHF/Llama-3-8b-rm-mixture` (Hu et al., 2024), which is trained from its corresponding SFT checkpoint. For `Mistral-7B`, we train a reward model from scratch using logsigmoid loss, as defined in Eq. 1, on the `Skywork/Skywork-Reward-Preference-80K-v0.1` (Liu & Zeng, 2024). For details on training procedures and hyperparameters, please refer to Appendix D.1.

**RM Calibration Dataset.** We employ a mixture of open-source datasets, and filter samples to ensure a high distinction between scores of chosen and rejected responses. Subsequently, we prepend the confidence-query system prompt shown in Fig 4 to each response. We then randomly assign high and low confidence scores to create four response types: chosen with high/low confidence and rejected with high/low confidence. Detailed information on dataset compositions is in Appendix C.3.

**RLHF Dataset.** We use a subset of `RLHFlow/prompt-collection-v0.1` (Dong et al., 2024) to accommodate computational resources. We randomly select 20,480 prompts and integrate a confidence-query system prompt into 25% of single-turn prompts to elicit verbalized confidence from the model, as exemplified in Figure 4. For clarity, we refer to the original 20,480 prompts as the **clean version** and those with the confidence-query system prompts added as the **modified version**.

**Evaluation Datasets.** We use six datasets for evaluation: GSM8K (Cobbe et al., 2021), CommonsenseQA (Talmor et al., 2019), SciQ (Welbl et al., 2017), ObjectCounting from BigBench (Srivastava et al., 2022), four Professional Knowledge datasets in MMLU (Hendrycks et al., 2020), and TruthfulQA (Lin et al., 2021). The datasets cover open-ended generation and multiple-choice questions.

**Compared Methods.** We compare our PPO-M and PPO-C against the following methods: (1) the SFT model, which serves as the initial checkpoint before RLHF training; (2) the PPO model, which employs a vanilla reward model during standard PPO training on the **clean version** dataset without

---

[3]These models are instruction-tuned and do not undergo the RLHF process.
[4]`https://github.com/OpenRLHF/OpenRLHF`

confidence-query system prompts; (3) PPO†, an ablation of PPO-M that includes confidence-query system prompts (**modified version**) during PPO training but still relies on the vanilla reward model.

## 4.2 MAIN RESULTS

**Both PPO-M and PPO-C consistently outperform other baselines across `Llama3-8B` and `Mistral-7B`.** In Table 1, we present the results of all five methods across six datasets. Compared to SFT, vanilla PPO shows a degradation in calibration (higher ECE and lower AUC) while generally improving accuracy. Among all methods, PPO-M and PPO-C consistently achieve lower ECE and higher AUC across both models and prompting strategies, highlighting their superior calibration ability. Furthermore, PPO-M and PPO-C maintain comparable or even higher accuracy, demonstrating that improved calibration does not come at the expense of model performance. Compared to PPO†, an ablation of PPO-M, PPO-M and PPO-C exhibit better calibration. This is because PPO†, while incorporating confidence-query system prompts during PPO training, still relies on the vanilla reward model instead of the calibrated reward model introduced in Sec. 3. This further indicates the importance of properly calibrating reward scores to mitigate bias toward high-confidence responses.

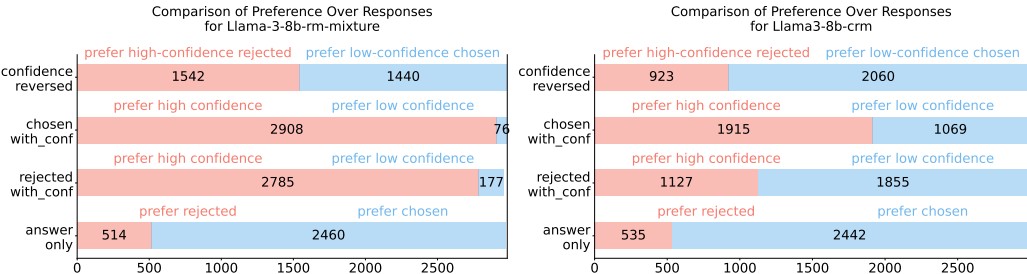

Figure 6: Preference distributions for `Llama3-8b-rm-mixture` (Pre-Calibrated Version) and `Llama3-8b-crm` (Post-Calibrated Version) on the modified `RewardBench` dataset across four modes: CONFIDENCE_REVERSED, CHOSEN_WITH_CONF, REJECTED_WITH_CONF, ANSWER_ONLY.

**Calibrated Reward Models.** Figure 6 illustrates the preference distributions of the calibrated reward model compared to the pre-calibrated version. The chosen and rejected ratio on the original responses without appended confidence scores (row 4) shows no significant difference between two models. However, when evaluated on rejected responses with high and low confidence scores (row 3), the pre-calibrated version consistently favors high-confidence responses. In contrast, the calibrated reward model demonstrates a preference for low-confidence responses – a behavior we aim to achieve.

## 5 ANALYSIS

In this section, we examine how our proposed methods influence the language model's abilities in instruction-following, and its engagement in conversational settings. Furthermore, we present how to extend our approach to Direct Preference Optimization (DPO) models and the results of the extension.

### 5.1 INSTRUCTION-FOLLOWING CAPABILITIES

**Dataset.** To evaluate whether PPO-M and PPO-C compromise the instruction-following abilities of LLMs gained through PPO, we assess their performance on two benchmarks: MT-Bench (Zheng et al., 2024) and Arena-Hard (Li et al., 2024). MT-Bench consists of 80 high-quality, multi-turn questions designed to evaluate LLMs across various aspects, while Arena-Hard contains 500 technical problem-solving queries and demonstrates a stronger agreement with human preference rankings.

**PPO-M and PPO-C do not compromise LLM instruction-following abilities.** Table 2 summarizes the average MT-Bench and Arena-Hard scores. As expected, PPO improves model performance compared to SFT. Additionally, models trained with PPO-M and PPO-C achieve scores comparable to or even slightly higher than those trained with standard PPO, highlighting that our calibration methods effectively preserve instruction-following abilities.

| Methods | | GSM8K | | | SciQ | | | CommonsenseQA | | |
|---|---|---|---|---|---|---|---|---|---|---|
| | | ECE ↓ | AUC ↑ | ACC ↑ | ECE ↓ | AUC ↑ | ACC ↑ | ECE ↓ | AUC ↑ | ACC ↑ |
| **Llama3-8B** | | | | | | | | | | |
| | SFT | 0.8608 | 0.5184 | 0.1221 | 0.0931 | 0.6067 | 0.873 | 0.2075 | 0.5889 | 0.7183 |
| DA | PPO | 0.8843 | 0.5021 | 0.1099 | 0.0683 | **0.6507** | **0.911** | 0.1729 | 0.5815 | 0.7641 |
| | PPO† | 0.8954 | 0.5 | 0.1046 | 0.0958 | 0.5047 | 0.904 | 0.2222 | 0.5113 | **0.7748** |
| | **PPO-M** | 0.8393 | **0.57** | **0.119** | **0.0267** | 0.6115 | 0.898 | 0.1206 | 0.5568 | 0.7707 |
| | **PPO-C** | **0.8025** | 0.5343 | 0.1046 | 0.0319 | 0.5892 | 0.906 | **0.0457** | **0.5835** | 0.7699 |
| | SFT | 0.4369 | 0.5138 | 0.5481 | 0.0944 | 0.65 | 0.856 | 0.1928 | 0.6155 | 0.7101 |
| CoT | PPO | 0.2566 | 0.5229 | 0.7392 | 0.0862 | **0.6763** | **0.879** | 0.1767 | **0.6287** | **0.7363** |
| | PPO† | 0.2553 | 0.5044 | 0.743 | 0.1265 | 0.5452 | 0.868 | 0.2654 | 0.5615 | 0.7191 |
| | **PPO-M** | 0.1909 | 0.5499 | **0.7703** | 0.0392 | 0.6635 | 0.877 | 0.1555 | 0.579 | 0.7346 |
| | **PPO-C** | **0.1546** | **0.5579** | 0.7635 | **0.0183** | 0.6473 | 0.868 | **0.1166** | 0.6049 | 0.7191 |
| **Mistral-7B** | | | | | | | | | | |
| | SFT | 0.8628 | 0.5747 | 0.0902 | 0.0952 | 0.5877 | 0.882 | 0.1634 | 0.56 | 0.774 |
| DA | PPO | 0.8675 | **0.583** | 0.097 | 0.0973 | 0.5497 | **0.89** | 0.1772 | 0.5594 | 0.7748 |
| | PPO† | 0.8851 | 0.5464 | 0.0877 | 0.1117 | 0.5439 | 0.885 | 0.1848 | **0.5674** | **0.7756** |
| | **PPO-M** | **0.7963** | 0.5055 | **0.1016** | **0.0108** | 0.5090 | 0.888 | **0.1163** | 0.5303 | 0.7625 |
| | **PPO-C** | 0.8161 | 0.534 | 0.0849 | 0.0399 | **0.5791** | 0.887 | 0.1311 | 0.5426 | 0.7592 |
| | SFT | 0.4124 | 0.5277 | 0.5785 | 0.1124 | 0.6238 | 0.872 | 0.1908 | 0.6205 | 0.7518 |
| CoT | PPO | 0.4146 | 0.5228 | 0.58 | 0.1126 | 0.5794 | 0.877 | 0.1867 | 0.6238 | 0.7699 |
| | PPO† | 0.3932 | 0.5096 | 0.6035 | 0.1044 | 0.5693 | 0.885 | 0.2056 | 0.6135 | 0.7518 |
| | **PPO-M** | **0.3379** | **0.5974** | 0.5982 | **0.0388** | 0.6584 | **0.886** | **0.1157** | 0.6118 | 0.7666 |
| | **PPO-C** | 0.377 | 0.5641 | **0.6065** | 0.0848 | **0.6951** | **0.886** | 0.1311 | **0.6367** | **0.774** |

| Methods | | TruthfulQA | | | Object Counting | | | Professional Knowledge | | |
|---|---|---|---|---|---|---|---|---|---|---|
| | | ECE ↓ | AUC ↑ | ACC ↑ | ECE ↓ | AUC ↑ | ACC ↑ | ECE ↓ | AUC ↑ | ACC ↑ |
| **Llama3-8B** | | | | | | | | | | |
| | SFT | 0.4613 | 0.5506 | 0.4113 | 0.5054 | 0.5212 | 0.483 | 0.4308 | 0.5175 | 0.4798 |
| DA | PPO | 0.425 | 0.5443 | 0.4651 | 0.508 | 0.4988 | 0.491 | 0.4078 | 0.4944 | **0.5046** |
| | PPO† | 0.5477 | 0.5246 | 0.4406 | 0.497 | 0.5 | 0.503 | 0.4951 | 0.4975 | 0.5009 |
| | **PPO-M** | 0.3991 | **0.5813** | **0.47** | 0.4789 | 0.5227 | 0.505 | 0.3848 | 0.4926 | 0.502 |
| | **PPO-C** | **0.3486** | 0.4856 | 0.4455 | **0.4405** | 0.5309 | 0.509 | **0.3318** | **0.5263** | 0.4798 |
| | SFT | 0.4436 | 0.5745 | 0.4174 | 0.4545 | 0.5102 | 0.54 | 0.4644 | 0.5571 | 0.4242 |
| CoT | PPO | 0.4726 | 0.5851 | 0.4113 | 0.3651 | 0.5023 | 0.634 | 0.4309 | 0.5606 | **0.4635** |
| | PPO† | 0.5535 | **0.5921** | 0.4076 | 0.337 | 0.5 | 0.663 | 0.5496 | 0.5219 | 0.4316 |
| | **PPO-M** | 0.4283 | 0.5674 | 0.437 | 0.2863 | **0.5341** | **0.703** | 0.4329 | 0.5422 | 0.4424 |
| | **PPO-C** | **0.3285** | 0.5193 | **0.4676** | **0.2525** | 0.5253 | 0.696 | **0.3798** | **0.5971** | 0.4353 |
| **Mistral-7B** | | | | | | | | | | |
| | SFT | 0.3307 | 0.5755 | 0.5704 | 0.5083 | 0.4989 | 0.491 | 0.4134 | 0.5018 | 0.5031 |
| DA | PPO | 0.3335 | 0.5567 | 0.5826 | 0.5008 | 0.5 | **0.499** | 0.4303 | 0.4889 | 0.4994 |
| | PPO† | 0.3233 | **0.5651** | 0.601 | 0.5119 | 0.499 | 0.488 | 0.4571 | 0.4919 | 0.4872 |
| | **PPO-M** | **0.245** | 0.5568 | **0.6071** | **0.4248** | 0.5067 | 0.483 | 0.3716 | 0.489 | 0.502 |
| | **PPO-C** | 0.2679 | 0.5456 | 0.5887 | 0.4947 | **0.5242** | 0.484 | **0.3693** | **0.51** | **0.505** |
| | SFT | 0.3657 | 0.6067 | 0.5398 | 0.4862 | 0.5072 | 0.512 | 0.4863 | 0.5369 | 0.4554 |
| CoT | PPO | 0.3677 | 0.5911 | 0.5581 | 0.4599 | 0.4991 | 0.54 | 0.4783 | 0.5275 | 0.4761 |
| | PPO† | 0.3657 | 0.6089 | **0.5594** | 0.455 | 0.5022 | 0.543 | 0.4735 | 0.5215 | **0.4865** |
| | **PPO-M** | **0.3142** | **0.6399** | 0.541 | **0.4134** | **0.5496** | 0.56 | **0.4090** | 0.5526 | 0.4579 |
| | **PPO-C** | 0.3213 | 0.6108 | 0.5545 | 0.4344 | 0.5095 | **0.563** | 0.4248 | **0.5588** | 0.4731 |

Table 1: Performance comparison of SFT, PPO, PPO†, PPO-M and PPO-C across six datasets using `Llama3-8B` and `Mistral-7B`. SFT denotes Supervised Fine-Tuned checkpoints, serving as the starting points for all methods. PPO† denotes an ablation of our PPO-M method which uses vanilla reward model in PPO training but on our modified dataset (with confidence-query system prompts).

In contrast, PPO† exhibits inferior performance compared to both PPO and our proposed methods. We hypothesize that this decline is primarily due to the reduced prompt diversity caused by the repetitive inclusion of confidence-query system prompts during training. To validate this hypothesis, we conduct additional experiments analyzing the impact a higher proportion of identical system prompts (See Appendix E.6). Notably, our analysis reveals that as the fraction of repeated system prompts increases, MT-Bench scores tend to decrease. These results consistently confirm a negative correlation the proportion of confidence-query system prompts used in training and model performance on MT-Bench.

Table 2: Results on MT-Bench and Arena-Hard.

| Model | Method | MT-Bench ↑ | Arena-Hard ↑ |
|---|---|---|---|
| | SFT | 7.34 | 10.0 |
| Llama3-8B | PPO | 8.00 | **14.6** |
| | PPO† | 7.81 | 13.4 |
| | **PPO-M** | **8.05** | 14.1 |
| | **PPO-C** | 7.87 | 13.7 |
| | SFT | 7.65 | 9.2 |
| Mistral-7B | PPO | 7.84 | 10.5 |
| | PPO† | 7.83 | **11.7** |
| | **PPO-M** | **7.95** | 9.9 |
| | **PPO-C** | 7.92 | 11.4 |

## 5.2 EXTENSION TO DPO

**Setup.** The CRM loss in Eq. 2, which calibrates the reward model using an augmented binary pairwise dataset, can naturally be extended to DPO training, as DPO models function as implicit reward models (Rafailov et al., 2024). We define this extension as Calibrated DPO (CDPO) in Eq. 4.

$$\mathcal{L}_{\text{CDPO}}(\pi_\theta; \pi_{\text{ref}}) = - \mathbb{E}_{(x,y_c,y_r,\hat{x},(y_c,h),(y_c,l),(y_r,h),(y_r,l))\sim\mathcal{D}} \left[\log\sigma(r(x,y_c) - r(x,y_r))\right.$$
$$\left. + w(\log\sigma(r(\hat{x},(y_c,h)) - r(\hat{x},(y_c,l))) + \log\sigma(r(\hat{x},(y_r,l)) - r(\hat{x},(y_r,h))))\right] \quad (4)$$

where $r(x,y) = \beta\log\frac{\pi_\theta(y|x)}{\pi_{\text{ref}}(y|x)}$ represents the implicit reward defined by model $\pi_\theta$ and its reference model $\pi_{\text{ref}}$. In this context, $(y_c, h)$ and $(y_r, l)$ denote the model responses paired with high and low confidence, respectively, with subscripts indicating whether it is a chosen or rejected response. $\hat{x}$ represents the prompt prepended with confidence-query system prompt. $w$ is the scaling coefficient.

The first term in Eq. 4 preserves the original DPO objective, preventing forgetting, since DPO models rely on subtle probability differences to effectively distinguish between chosen and rejected responses.

We use the `Mistral-7B` DPO version (*i.e.*, `teknium/OpenHermes-2.5-Mistral-7B` as the reference model and `NousResearch/Nous-Hermes-2-Mistral-7B-DPO` as the DPO version) for the experiment. We fine-tune the DPO model on our RM calibration Dataset using Eq. 4.

**Results.** As shown in Table 4, CDPO significantly improves model calibration across all six datasets, achieving consistently lower ECE and higher AUC compared to other methods. Notably, CDPO reduces ECE by over 50% on TruthfulQA, CommonsenseQA, and Professional Knowledge datasets. Although a slight decline in performance is observed between CDPO and DPO†, CDPO still achieves performance comparable to the original DPO checkpoint, af-

| Model | Method | MT-Bench ↑ | Arena-Hard ↑ |
|---|---|---|---|
| Mistral-7B | SFT | 7.65 | 9.2 |
| | DPO | 7.83 | 13.4 |
| | DPO† | 7.83 | 14.3 |
| | CDPO | **7.85** | **15.9** |

Table 3: Comparison of DPO and CDPO on MT-Bench And Arena-Hard scores for `Mistral-7B`.

firming that calibration does not compromise overall model capabilities. Results on MT-Bench and Arena-Hard are presented in Table 3. For `Mistral-7B`, training on additional data improves both MT-Bench and Arena-Hard scores, and CDPO further amplifies these gains compared to standard DPO on the calibration dataset (DPO†). Results for `Llama3-8B` are provided in Appendix E.11.

## 6 RELATED WORKS

**LLM Calibration.** Model Calibration aims to align a model's confidence with its accuracy. Recent studies show that LLMs often exhibit overconfidence (Tian et al., 2023; Chen et al., 2024a; Xiong et al., 2023; Achiam et al., 2023). Previous studies have explored methods such as scaling-based (Deng et al., 2023; Guo et al., 2017; Zhang et al., 2020) approaches and nonparametric methods, such as binning (Zadrozny & Elkan, 2001). Recent work has introduced verbalized confidence (Lin et al.,

| Methods | | GSM8K | | | SciQ | | | CommonsenseQA | | |
|---|---|---|---|---|---|---|---|---|---|---|
| | | ECE ↓ | AUC ↑ | ACC ↑ | ECE ↓ | AUC ↑ | ACC ↑ | ECE ↓ | AUC ↑ | ACC ↑ |
| DA | SFT | 0.8628 | 0.5747 | 0.0902 | 0.0952 | 0.5877 | 0.882 | 0.1634 | 0.56 | 0.774 |
| | DPO | 0.8704 | 0.5916 | 0.0887 | 0.0845 | 0.581 | 0.892 | 0.177 | 0.5744 | 0.7682 |
| | DPO† | 0.8057 | 0.5409 | **0.0826** | **0.0149** | 0.5215 | 0.884 | 0.1157 | 0.5491 | **0.7772** |
| | CDPO | **0.6767** | **0.6163** | 0.0781 | 0.0967 | **0.7236** | **0.89** | **0.0513** | **0.6165** | 0.7666 |
| CoT | SFT | 0.4124 | 0.5277 | 0.5785 | 0.1124 | 0.6238 | 0.872 | 0.1908 | 0.6205 | 0.7518 |
| | DPO | 0.4184 | 0.5253 | 0.5716 | 0.094 | 0.5837 | 0.896 | 0.1849 | 0.6145 | 0.7625 |
| | DPO† | 0.3456 | 0.5953 | 0.5989 | **0.0214** | 0.6687 | **0.898** | 0.0916 | **0.6553** | **0.7764** |
| | CDPO | **0.1889** | **0.7178** | **0.6164** | 0.0553 | **0.7623** | 0.883 | **0.0676** | 0.6498 | 0.7633 |

| Methods | | TruthfulQA | | | Object Counting | | | Professional Knowledge | | |
|---|---|---|---|---|---|---|---|---|---|---|
| | | ECE ↓ | AUC ↑ | ACC ↑ | ECE ↓ | AUC ↑ | ACC ↑ | ECE ↓ | AUC ↑ | ACC ↑ |
| DA | SFT | 0.3307 | 0.5755 | 0.5704 | 0.5083 | 0.4989 | 0.491 | 0.4134 | 0.5018 | 0.5031 |
| | DPO | 0.2912 | 0.5725 | 0.6181 | 0.5149 | 0.501 | 0.485 | 0.4321 | 0.4967 | 0.4913 |
| | DPO† | 0.2124 | 0.5674 | 0.6487 | 0.4336 | **0.5436** | 0.485 | 0.3649 | 0.5208 | **0.5091** |
| | CDPO | **0.104** | **0.6225** | **0.661** | **0.3955** | 0.5304 | **0.491** | **0.2574** | **0.5451** | 0.4972 |
| CoT | SFT | 0.3657 | 0.6067 | 0.5398 | 0.4862 | 0.5072 | 0.5120 | 0.4863 | 0.5369 | 0.4554 |
| | DPO | 0.3251 | 0.629 | 0.6022 | 0.4581 | 0.5003 | 0.5430 | 0.4950 | 0.5314 | 0.4609 |
| | DPO† | 0.2169 | 0.6176 | **0.6377** | 0.4037 | **0.5585** | 0.539 | 0.3679 | 0.5587 | **0.4961** |
| | CDPO | **0.1756** | **0.685** | 0.6193 | **0.322** | 0.5139 | **0.553** | **0.2917** | **0.614** | 0.4817 |

Table 4: Performance comparison of SFT, DPO, DPO†, and CDPO across six datasets using `Mistral-7B`. SFT and DPO denote the reference and trained DPO models, respectively. DPO† and CDPO initiate from the trained DPO checkpoint; DPO† applies standard DPO on the calibration dataset, focusing on chosen and rejected pairs to assess the impact of training with additional data.

2022), where models are prompted to directly output confidence scores. Most studies focus on pre-trained and instruction-tuned LLMs (Lin et al., 2022; Han et al., 2024), while other studies examine RLHF-trained LLMs, proposing calibration through prompting strategies (Xiong et al., 2023; Tian et al., 2023). More recent work leverages Reinforcement Learning for calibration (Xu et al., 2024; Tao et al., 2024), which aligns closely with our study. Our study contributes by identifying the potential cause for overconfidence in RLHF-LLMs and proposing calibration of the reward models or reward score calculations to be seamlessly integrated into the existing PPO framework. In addition, our approach does not compromise the model's generalization capabilities in open-ended generation.

**LLM Alignment And Reward Modeling.** Reinforcement Learning from Human Feedback (RLHF) (Ouyang et al., 2022; Christiano et al., 2017; Bai et al., 2022) has been widely applied to align LLMs with human preferences. This pipeline typically involves Supervised Fine-Tuning (SFT), reward modeling, and policy optimization using Proximal Policy Optimization (PPO) (Schulman et al., 2017). Recent studies have explored variations of this pipeline to address noisy human preferences (Hong et al., 2022; Wang et al., 2024a) and to improve training efficiency by eliminating the need for a separate reward model with Direct Preference Optimization (Rafailov et al., 2024).

A comprehensive discussion of related works, including detailed analysis, is provided in Appendix A.

## 7 CONCLUSION

This paper addresses the issue of overconfidence in RLHF-LLMs by identifying a systematic bias in reward models that favors high-confidence responses, regardless of their actual quality. To mitigate this bias, we propose PPO-M, which calibrates reward modeling by aligning confidence levels with response quality, and PPO-C, which adjusts standard reward model scores during PPO training. Both methods integrate seamlessly into the RLHF framework. Extensive experiments on various benchmarks demonstrate the effectiveness of our approaches in reducing expected calibration error while maintaining accuracy and robust instruction-following capabilities in open-ended generation.

ACKNOWLEDGMENT

We thank the anonymous reviewers for their valuable and insightful feedback, which contributed to improving this work. This research was supported in part by the NVIDIA Academic Grant Program.

REPRODUCIBILITY STATEMENT

To facilitate reproducibility, we provide detailed information on the datasets used (see Appendix C), implementation details (see Appendix D), and supplementary results and analysis (see Appendix E).

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

# A   RELATED WORKS

**LLM Calibration.**   Model Calibration aims to align a model's confidence with its accuracy. It has been observed that modern neural networks, including Large Language Models (LLMs), often exhibit overconfidence, suggesting poor calibration (Tian et al., 2023; Chen et al., 2024a; Xiong et al., 2023; Achiam et al., 2023). Previous studies have explored methods like scaling-based (Deng et al., 2023; Guo et al., 2017; Zhang et al., 2020) approaches and nonparametric methods such as binning (Zadrozny & Elkan, 2001). Among these, temperature scaling (Guo et al., 2017; Zhang et al., 2020) has been proven to be effective when combined with large pre-trained LLMs (Kadavath et al., 2022; Xiao et al., 2022; Kuhn et al., 2023). However, previous evaluations focus on probabilities derived from model logits (Hendrycks et al., 2020; Mukhoti et al., 2020; Guo et al., 2017; Minderer et al., 2021), which can sometimes be inaccessible in proprietary models and unclear to human users.

Recently, verbalized confidence has been introduced (Lin et al., 2022), prompting models to directly output confidence scores alongside responses. While most studies focus on calibrating pre-trained LLMs through supervised fine-tuning (Lin et al., 2022; Han et al., 2024), which typically involves sampling responses and calculating average accuracy as the estimation for ground truth confidence scores, other studies have examined verbalized confidence in instruction fine-tuned and RLHF-trained LLMs, and propose calibration through prompting strategies (Xiong et al., 2023; Tian et al., 2023).

More recent work leverages Reinforcement Learning for calibration (Xu et al., 2024; Tao et al., 2024) which closely aligns with the focus of our study. We contribute by identifying a potential cause of overconfidence in RLHF-trained LLMs and proposing calibration of the reward models or reward score calculations to mitigate this issue. The proposed methods can be seamlessly integrated into the existing PPO framework. Unlike supervised fine-tuning (SFT) methods, which require datasets with ground truth labels for accuracy calculation – limiting their applicability to open-ended generation tasks – our approach does not compromise the model's generalization capabilities in such settings.

**LLM Alignment And Reward Modeling.**   Reinforcement Learning from Human Feedback (RLHF) (Ouyang et al., 2022; Christiano et al., 2017; Bai et al., 2022) has been widely applied to align LLMs with human preferences. This pipeline typically comprises three steps: Supervised Fine-Tuning (SFT), the collection of pairwise ranking data and the development of a reward model, and optimization of the policy model obtained from the first step using Proximal Policy Optimization (PPO) (Schulman et al., 2017). The effectiveness of PPO depends heavily on the accuracy and robustness of the reward model. Following traditional Bradley-Terry reward models (Bradley & Terry, 1952), training typically utilizes a binary pairwise dataset. However, human-labeled preferences are often noisy or exhibit conflicting signals (Hong et al., 2022; Knox et al., 2022; Wang et al., 2024a). To address these challenges, several methods have been proposed, including introducing a margin to guide the reward model in assigning greater weight to more distinguishable comparison pairs (Touvron et al., 2023; Wang et al., 2024a), and employing multi-objective reward modeling that considers joint preference, such as "helpfulness, correctness, coherence". (Dong et al., 2023b; Zhou et al., 2023; Wang et al., 2024b; Chen et al., 2024b; Chakraborty et al., 2024; Wang et al., 2024c).

Although the RLHF pipeline has proven effective in aligning LLMs with human preferences, Proximal Policy Optimization (PPO) presents several challenges, including reward hacking, sensitivity to hyperparameters, and substantial computational resource demands, which complicate its implementation and practical use. To address these challenges, various alternatives have been proposed (Dong et al., 2023a; Yuan et al., 2023; Zhao et al., 2023; Rafailov et al., 2024; Song et al., 2023b; Azar et al., 2023; Ethayarajh et al., 2024; Hong et al., 2024; Liu et al., 2024; Meng et al., 2024). Among these, Direct Preference Optimization (DPO) has gained significant adoption (Rafailov et al., 2024; Dubey et al., 2024). DPO defines the preference loss as a direct function of the policy model, thereby eliminating the need for a separate reward model. However, despite these advancements, limited research has examined how reward models contribute to the confidence calibration of LLMs. In this study, we address this gap by highlighting the vulnerability of reward models trained through different approaches, which can be easily biased by simply adding confidence scores. Furthermore, we propose two methods to calibrate these models and effectively reduce overconfidence in RLHF-trained LLMs.

## B LIMITATION AND BROADER IMPACT

### B.1 LIMITATION

While we demonstrate that directly applying CRM loss 2 to DPO training can effectively reduce ECE and increase AUC, thereby improving model calibration, this method is not explicitly optimized for DPO. Our observations indicate some degree of performance degradation, highlighting the need for future work to explore hyperparameter tuning or development of a more specifically designed dataset.

### B.2 BROADER IMPACT

Our work emphasizes model calibration, offering two methods that can be applied across a wide range of domains requiring well-calibrated language models. Improved model calibration enhances the reliability, trustworthiness, and safety of general AI systems, thereby benefiting the communities.

## C DATASETS

In this section, we provide detailed descriptions of the datasets utilized in this study, including those used for the preliminary experiments, reward modeling, reward model calibration, and PPO training.

### C.1 PRELIMINARY EXPERIMENTS DATASET

In Section 2.2, we demonstrate the inherent bias in reward models toward responses with high confidence scores by modifying `allenai/reward-bench` dataset. This modification involves prepending a confidence-query system prompt and appending random confidence scores to the original model responses. For this purpose, we assign scores ranging from 0 to 3 to indicate low confidence and 7 to 10 to signify high confidence. Figure 7 provides an example from the modified dataset on REJECTED_WITH_CONF, showing a rejected response with appended low confidence scores.

```
Modified Reward Bench Example

System Prompts:
For the following question, provide your best response first, followed by your
↪  confidence in the accuracy or helpfulness of your response. Rate your confidence
↪  on a scale from 0 to 10.
```Example Format:
<Your responses>
Confidence: <Insert your numerical confidence level from 0 to 10, reflecting how
↪  certain you are that your answer is accurate or helpful.>```

Ensure that your response strictly adheres to this format. Explicitly include the word
↪  'Confidence:' in your response.
────────────────────────────────────────────────────────────
User Prompt:
How do I detail a car?
────────────────────────────────────────────────────────────
Model Response:
Detailing a car involves washing the exterior and interior of the car, as well as
↪  polishing and waxing the exterior. Interior detailing typically involves
↪  vacuuming, cleaning the upholstery and air vents, polishing the dashboard and
↪  console, and dusting. Polishing and waxing the exterior will depend on the
↪  condition of the paint, but typically involves applying a polish and wax to make
↪  it shine.
Confidence: 2.
```

Figure 7: An example from the Modified `RewardBench` in mode: REJECTED_WITH_CONF.

## C.2 REWARD MODEL TRAINING DATASETS

For `Mistral-7B`, we utilize `Skywork/Skywork-Reward-Preference-80K-v0.1` (Liu & Zeng, 2024), an open-source pairwise binary dataset and train the reward model from scratch.

## C.3 REWARD MODEL CALIBRATION DATASETS.

In order to compile the dataset for calibrating reward models, we filter samples from multiple open-source datasets. Table 5 lists the datasets utilized and the thresholds applied for each in detail.

Initially, we filter out samples that are multi-turn or have a tokenized length exceeding 8192, as multi-turn formats are unsuitable for assigning confidence scores, and truncation should be avoided. The threshold represents the preference strength (Wang et al., 2024a), defined as the difference between chosen and rejected scores. In datasets such as `RLHFlow/Argilla-Math-DPO-standard`, a preference strength below 1 often indicates that both chosen and rejected responses yield the same answer via different reasoning paths. Our objective is to calibrate the reward model to assign higher scores to high-confidence chosen responses and lower scores to high-confidence rejected responses, while reversing this pattern for low-confidence responses. However, when both responses produce the same mathematical solution through different reasoning, it is inappropriate and misleading for low-confidence rejected responses to receive higher scores. Consequently, we exclude these ambiguous samples and retain only those with a significant discrepancy between chosen and rejected responses. To balance computational resources, we set a threshold to retain approximately 2,500 samples per dataset. For datasets lacking specific chosen and rejected scores, we randomly select 2,500 samples.

| Dataset | Threshold |
|---|---|
| argilla/distilabel-capybara-dpo-7k-binarized (Daniele & Suphavadeeprasit, 2023) | 1 |
| RLHFlow/CodeUltraFeedback-standard (Weyssow et al., 2024) | 3 |
| argilla/ultrafeedback-binarized-preferences-cleaned (Bartolome et al., 2023) | 3.5 |
| RLHFlow/Helpsteer-preference-standard (Wang et al., 2023) | 2.5 |
| RLHFlow/Helpsteer2-standard (Wang et al., 2024d) | 2 |
| RLHFlow/Orca-distibalel-standard (Lian et al., 2023) | 2.0 |
| RLHFlow/SHP-standard (Ethayarajh et al., 2022) | 50 |
| RLHFlow/HH-RLHF-Helpful-standard (Bai et al., 2022) | NA |
| RLHFlow/Argilla-Math-DPO-standard | 1 |
| RLHFlow/PKU-SafeRLHF-30K-standard (Ji et al., 2024) | NA |
| CyberNative/Code_Vulnerability_Security_DPO | NA |
| fblgit/simple-math-DPO (Murias, 2024) | NA |

Table 5: Dataset compositions.

## C.4 PPO DATASETS

For PPO training, we filter out prompts with a tokenized length exceeding 8192 to prevent truncation and randomly select 20,480 prompts from `RLHFlow/prompt-collection-v0.1` (Dong et al., 2024). To elicit verbalized confidence from the model, we integrate a confidence-query system prompt into single-turn prompts. The system prompt is included in 25% of the single-turn prompts for main results. Figure 8 illustrates an example from the dataset that incorporates this system prompt.

---

**PPO Prompts Example**

```
System Prompts:
For the following question, provide your best response first, followed by your
↪   confidence in the accuracy or helpfulness of your response. Rate your confidence
↪   on a scale from 0 to 10.
```Example Format:
<Your generated response>
Confidence: <Insert your numerical confidence level from 0 to 10, reflecting how
↪   certain you are that your answer is accurate or helpful.>```

Ensure that your response strictly adheres to this format. Explicitly include the word
↪   'Confidence:' in your response.
to the left if the cell is full
________________________________________________________________

User Prompt:
Write me an excel function to sum up the values in the cells in a column to the left
↪   if the cell is full
```

Figure 8: PPO Prompt Example.

## C.5 EVALUATION DATASETS.

We examine six datasets encompassing six distinct categories: **Arithmetic Reasoning**, **Common-sense Knowledge**, **Symbolic Reasoning**, **Truthful Reasoning**, and **Professional Knowledge** Collectively, these datasets include a mix of open-ended generation tasks and multiple-choice questions.

- **GSM8K (Cobbe et al., 2021):** This dataset contains high-quality, linguistically diverse grade school math word problems. We utilize the test split, which contains 1319 samples.

- **CommonsenseQA (Talmor et al., 2019):** This dataset features a multiple-choice question format requiring commonsense knowledge. We use the test split, containing 1,221 samples.

- **TruthfulQA (Lin et al., 2021):**[5] This dataset contains 817 questions designed to test whether the model can generate truthful responses while recognizing false beliefs and misconceptions. We utilize the multiple-choice format of the dataset and consider one single target answer. To ensure the correct label is not predictably the first option, we randomly shuffle the answer options and corresponding true labels. We format the questions as lettered multiple-choices and instruct the model to select the best answer from the options provided.

- **SciQ (Welbl et al., 2017) :** This dataset contains crowdsourced science exams. We use the test split for evaluation, which includes 1000 examples. It is a multiple-choice dataset, with each question offering four answer options. Similar to TruthfulQA and CommonsenseQA, we assign a letter to each answer option and request the model to output the answer letter.

- **Object Counting in BigBench (Srivastava et al., 2022):** BigBench is a collaborative benchmark encompassing over 200 tasks. For Symbolic Reasoning, we focus on one subset, Object Counting, which includes 1000 samples. This open-ended generation task evaluates whether models can accurately determine the number of objects mentioned in the questions.

- **Professional Knowledge in MMLU (Hendrycks et al., 2020):** MMLU is a multitask benchmark that includes multiple-choice format questions from diverse knowledge domains. For the Professional Knowledge category, we combine the test sets from four subsets: Professional Accounting, Professional Law, Professional Medicine, and Professional Teaching.

## D IMPLEMENTATION DETAILS

In this section, we describe the implementation details for all experiments.

---

[5]https://huggingface.co/datasets/truthfulqa/truthful_qa/viewer/
multiple_choice

## D.1 REWARD MODEL TRAINING

This study utilizes two reward models. For `Llama3-8B`, we use an off-the-shelf checkpoint from `OpenRLHF/Llama3-8b-rm-mixture`. For `Mistral-7B`, the reward model is trained from scratch using `teknium/OpenHermes-2.5-Mistral-7B`, referred to as `Mistral-7B-RM`.

### D.1.1 HYPERPARAMETERS

| Parameter | Mistral-7B |
|---|---|
| Train Batch Size | 512 |
| Micro Batch Size | 1 |
| Learning Rate | 2e-6 |
| Max Length | 8192 |
| LR Scheduler | cosine_with_min_lr |
| Warmup Ratio | 0.03 |
| Optimizer | AdamW |
| Weight Decay | 0.01 |
| Epochs | 2 |

Table 6: Hyperparameters for Reward Modeling.

We list the detailed hyperparameters used for training `Mistral-7B-RM` in Table 6 for reference.

## D.2 REWARD MODEL CALIBRATION

As stated in Section 3, we assume that reward models used for calibration are already trained beforehand and generally perform well. To this end, we utilize trained RM checkpoints, `OpenRLHF/Llama3-8b-rm-mixture` and `Mistral-7B-RM` for calibration. The calibrated versions of these models are referred to as `Llama3-8b-crm` and `Mistral-7B-crm`, respectively.

### D.2.1 HYPERPARAMETERS

Hyperparameters for calibrating `Llama3-8b-crm` and `Mistral-7B-RM` are provided in Table 7.

| Parameter | Llama3-8B-crm | Mistral-7B-crm |
|---|---|---|
| Train Batch Size | 256 | 256 |
| Micro Batch Size | 1 | 1 |
| Learning Rate | 9e-6 | 5e-6 |
| Max Length | 8192 | 8192 |
| LR Scheduler | cosine_with_min_lr | cosine_with_min_lr |
| Warmup Ratio | 0.03 | 0.03 |
| Optimizer | Adam | Adam |
| Epochs | 1 | 2 |

Table 7: Hyperparameters for Calibrating `Llama3-8B-crm` and `Mistral-7B-crm`.

## D.3 PPO TRAINING

Following the standard RLHF pipeline, we initialize the policy model using corresponding supervised fine-tuning checkpoints: `OpenRLHF/Llama3-8b-sft-mixture` for `Llama3-8B`, and `teknium/OpenHermes-2.5-Mistral-7B` for `Mistral-7B`. For standard PPO and PPO-C, we utilize the pre-calibrated reward models, specifically `OpenRLHF/Llama3-8b-rm-mixture` and `Mistral-7B-RM`. In standard PPO, the reward score is obtained on EOS token of the sequence.

For PPO-C, we apply our proposed calibrated reward calculation method (see Section 3 for details).

For PPO-M, we leverage `Llama3-8b-crm` and `Mistral-7B-crm` to calculate reward scores.

### D.3.1 HYPERPARAMETERS

For each model (`Llama3-8B` and `Mistral-7B`), we employ a consistent set of hyperparameters across PPO, PPO-M, and PPO-C to ensure fair comparisons and reproducibility, as detailed in Table 8.

| Parameter | Llama3-8B | Mistral-7B |
|---|---|---|
| Train Batch Size | 64 | 64 |
| Micro Batch Size | 2 | 2 |
| Micro Rollout Batch Size | 4 | 4 |
| Rollout Batch Size | 512 | 512 |
| Prompt Max Len | 1024 | 1024 |
| Generate Max Len | 1024 | 1024 |
| Actor Learning Rate | 5e-7 | 1e-7 |
| Critic Learning Rate | 9e-6 | 1e-6 |
| Actor Weight Decay | 0.0 | 0.01 |
| Critic Weight Decay | 0.0 | 0.0 |
| Initial KL Confidence | 0.01 | 0.05 |
| LR Scheduler | cosine_with_min_lr | cosine_with_min_lr |
| Warmup Ratio | 0.03 | 0.03 |
| Optimizer | Adam | Adam |
| Epochs | 1 | 1 |

Table 8: Hyperparameters for PPO Training.

### D.4 DPO TRAINING

In Section 5.2, we extend calibrated reward modeling (PPO-M) to DPO training using Eq. 4. Following the approach used for calibrating reward models, we leverage pre-trained DPO checkpoints.

For `Llama3-8B`, we utilize `princeton-nlp/Llama-3-Base-8B-SFT-DPO` as the DPO checkpoint and `princeton-nlp/Llama-3-8B-Base-SFT` as the reference model. For `Mistral-7B`, we use `NousResearch/Nous-Hermes-2-Mistral-7B-DPO` as the DPO checkpoint, with `teknium/OpenHermes-2.5-Mistral-7B` serving as the reference model.

### D.4.1 HYPERPARAMETERS

We list the hyperparameters used for DPO training `Nous-Hermes-2-Mistral-7B-DPO` and `Llama-3-Base-8B-SFT-DPO` in Table 9. The same set of hyperparameters is applied to both DPO and CDPO. However, it is important to note that the scaling coefficient $w$ is not utilized in DPO.

### D.5 EVALUATION AND PARSING

In this section, we provide a detailed overview of the generation configuration, prompting and parsing strategies. All evaluations are performed on a single Nvidia A100 80GB GPU with a batch size of 8.

### D.5.1 GENERATION CONFIGURATION

We use consistent settings for both preliminary and main experiments: temperature at 1.0, top-p at 1.0, top-k at 50, with a maximum token limit of 16 for direct answers and 256 for zero-shot CoT.

| Parameter | Llama3-8B | Mistral-7B |
|---|---|---|
| Train Batch Size | 128 | 128 |
| Micro Batch Size | 1 | 1 |
| Max Length | 4096 | 4096 |
| Learning Rate | 3e-7 | 3e-7 |
| Beta | 0.01 | 0.01 |
| Weight Decay | 0.0 | 0.0 |
| LR Scheduler | cosine_with_min_lr | cosine_with_min_lr |
| Warmup Ratio | 0.03 | 0.03 |
| Optimizer | Adam | Adam |
| Epochs | 1 | 1 |
| Zero Stage | 3 | 2 |
| Adam Offload | True | False |
| $w$ (scaling coefficient) | 1.0 | 0.5 |

Table 9: Hyperparameters for DPO and CDPO Training.

## D.6 EVALUATION PROMPTS

Following the format described in Tian et al. (2023), we modify the prompt to improve clarity and simplify the interpretation of the results. We consider two prompting strategies for evaluation: Direct Answer and Zero-Shot CoT (Kojima et al., 2022). The exact prompt is shown in Fig 9 and Fig 10, which also include a model response from GSM8K. For `answer_type`: we use `option letter` for multiple-choice questions and `number` for open-ended math problems. For `demo`: we use `(A)` for multiple-choice questions and `1` for open-ended math problems. Prompt formatting leverages the chat template in the tokenizer. Instructions are placed in the system prompt, and the question is placed in the user prompt. For models like Tulu-2 (Ivison et al., 2023), which lacks a system prompt section in the tokenizer chat template, we append the question after the instruction as the user prompt.

---

**Prompt for Direct Answers**

```
System Prompts:
For the following question, provide your answer including only the {answer_type}
↪   first, followed by your confidence in the accuracy or helpfulness of your
↪   response. Rate your confidence on a scale from 0 to 10.
Please respond only with your answer and a numerical confidence score. Do not include
↪   any additional text, characters, or explanations. Use the format demonstrated
↪   below for your response.
```Example Format:
Answer: <Insert only the {answer_type} here (e.g., {demo})>
Confidence: <Insert your numerical confidence level from 0 to 10, reflecting how
↪   certain you are that your answer is correct.>```

Ensure that your response strictly adheres to this format and contain only the
↪   {answer_type} and the confidence score. Explicitly include the words 'Answer:' and
↪   'Confidence:' in your response.
```
---
```
User Prompt:
Question: James decides to run 3 sprints 3 times a week.  He runs 60 meters each
↪   sprint.  How many total meters does he run a week?
```
---
```
Model Output:
Answer: 540
Confidence: 10
```

Figure 9: Direct Answers Prompt.

---

**Prompt for Zero-Shot CoT**

```
System Prompts:
For the following question, provide a step-by-step explanation of your thought process
↪   first, then offer your best answer and rate your confidence in the accuracy or
↪   helpfulness of each response on a scale from 0 to 10.
Use the format demonstrated below for your response.
```Example Format:
Explanation: <Your detailed explanation here, outlining how you arrived at your
↪   answer.>
Answer: <Insert your concise answer here, which should include a {answer_type} (e.g.,
↪   {demo})>
Confidence: <Insert your numerical confidence level from 0 to 10, reflecting how
↪   certain you are that your answer is correct.>```

Ensure that your response strictly adheres to this format. Explicitly include the
↪   words 'Explanation:', 'Answer:', and 'Confidence:' in your response.
─────────────────────────────────────────────────────────────────────────────
User Prompt:
Question: James decides to run 3 sprints 3 times a week.  He runs 60 meters each
↪   sprint.  How many total meters does he run a week?
─────────────────────────────────────────────────────────────────────────────
Model Output:
Explanation: James runs 3 sprints of 60 meters each, and he does this 3 times a week.
↪   Therefore, he runs 3 sprints * 60 meters per sprint = 180 meters per session. Since
↪   he does this 3 times a week, he runs a total of 180 meters/session * 3 sessions =
↪   540 meters in a week.
Answer: 540 meters
Confidence: 10
```

Figure 10: Zero-Shot CoT Prompt.

## D.7 PARSING DETAILS

**Regex Parsing Details.** To parse the confidence score from model-generated responses, we implement a stopping criterion that triggers only when numeric digits directly follow the phrase "Confidence: ". For responses that are initially unparseable, we set a retry limit up to ten attempts. If parsing failures persist, we manually append "Confidence: " to the model's response and resubmit it for completion, allowing the model to generate the missing score. This approach enables us to achieve nearly 100% success in parsing all responses. In the rare instances where parsing ultimately fails, we use an empty string as the default answer. Instead of assigning an arbitrary confidence score of 5.0 – which could introduce bias and artificially inflate ECE – we use the most frequently observed confidence score from successfully parsed responses as the default value. This approach ensures the assigned score could remain representative of the model's behavior and minimizes the risk of bias.

---

**GPT-4o Evaluation Prompt**

```
System Prompt:
You are a specialized evaluator designed to assess model responses against golden
↪   answers for various tasks and extract model confidence. Output your evaluation in
↪   JSON format.
─────────────────────────────────────────────────────────────────────────────
User Prompt:
Evaluate the semantic equivalence between the given model response and the provided
↪   golden answer. Determine if they convey the same meaning.
If the model response accurately matches the golden answer (i.e., the model response
↪   is correct), assign a score of 1. If the model response does not match the golden
↪   answer, assign a score of 0.
Additionally, extract the confidence score from the model response. If the model
↪   response does not explicitly state a confidence score, return -100.
Provide your answer in the following JSON format: {'correctness': 1 or 0,
↪   'confidence': X.X}
```

Figure 11: Prompts for GPT4-o Evaluation.

**GPT-4o Evaluation Details.** We use `gpt-4o-2024-08-06` to evaluate zero-shot CoT results. Leveraging GPT's structured output feature, we configure the model to generate results in JSON format, enabling straightforward and efficient parsing. The prompt used for this is shown in Figure 11.

# E  MORE RESULTS AND ANALYSIS

## E.1  OVERCONFIDENCE IN RLHF-LLMS

In this section, we present additional results from our preliminary experiments, demonstrating overconfidence in RLHF-trained LLMs across five datasets, as shown in Figure 12 to 16. These results show that RLHF-trained LLMs consistently exhibit verbalized overconfidence across datasets.

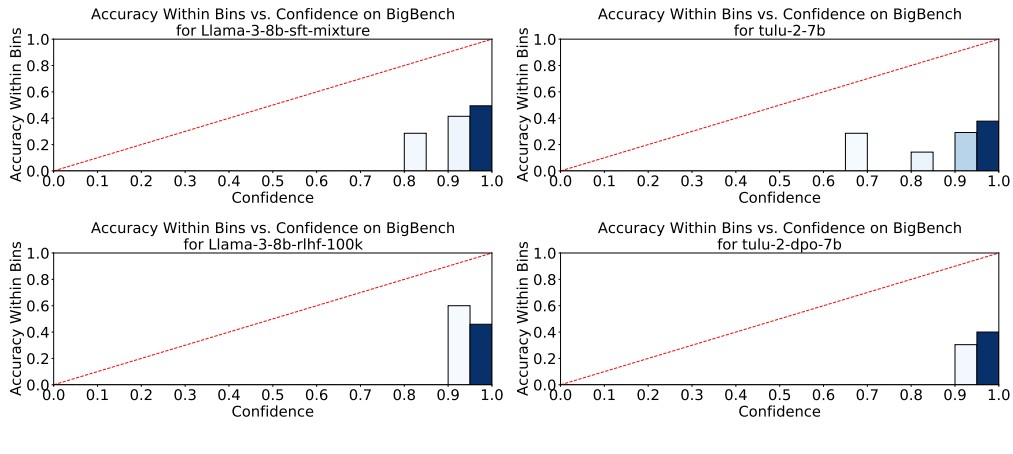

Figure 12: Confidence distributions of models on ObjectCount before (top) and after (bottom) RLHF.

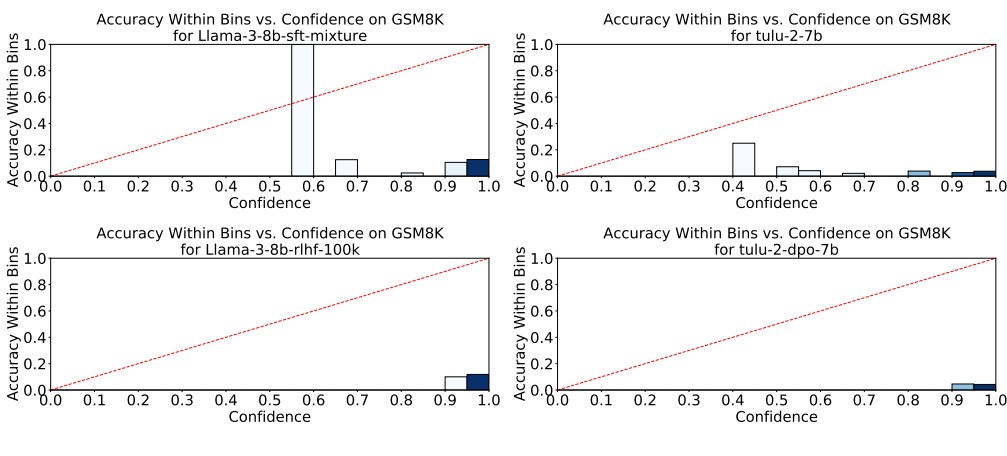

Figure 13: Confidence distributions of models on GSM8K before (top) and after (bottom) RLHF.

## E.2  REWARD MODELS ARE BIASED TOWARD HIGH CONFIDENCE SCORES

Following Section 2.2, we present additional results to further substantiate the observed phenomenon.

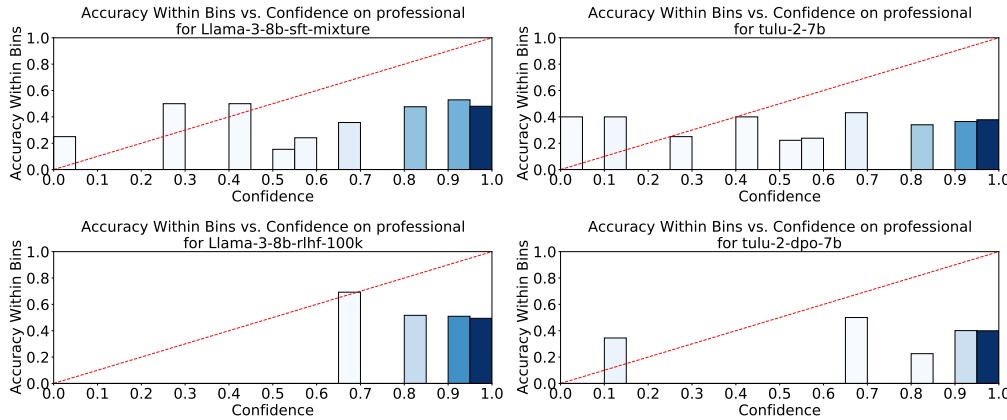

Llama3-8B-SFT and Llama3-8B-PPO;   Tulu-2-7B and Tulu-2-DPO-7B

Figure 14: Confidence distributions of models on Prof.Knowl before (top) and after (bottom) RLHF.

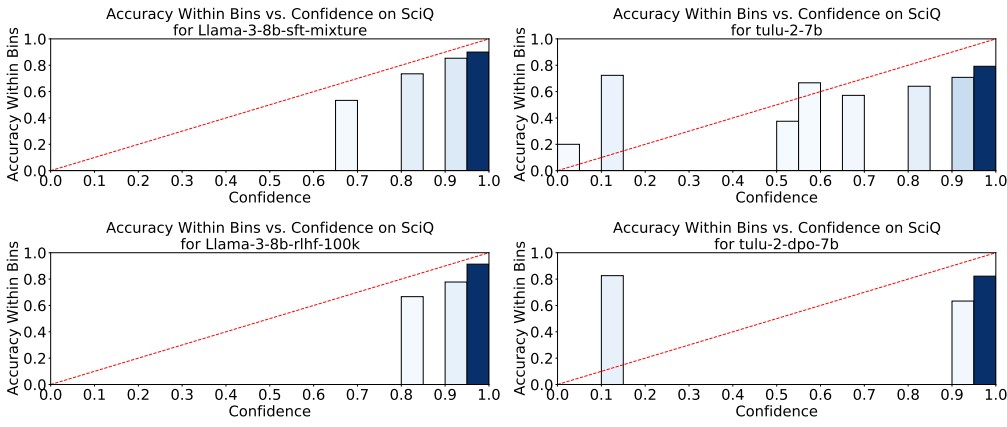

Llama3-8B-SFT and Llama3-8B-PPO;   Tulu-2-7B and Tulu-2-DPO-7B

Figure 15: Confidence distributions of models on SciQ before (top) and after (bottom) RLHF.

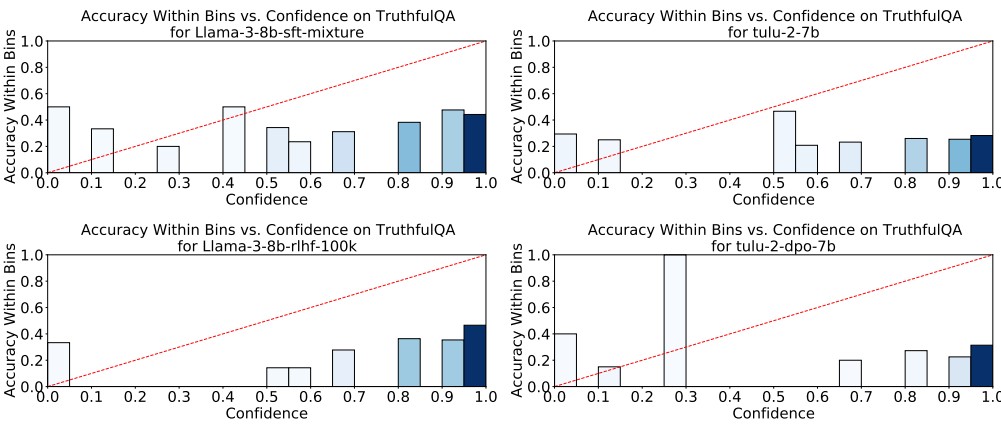

Llama3-8B-SFT and Llama3-8B-PPO;   Tulu-2-7B and Tulu-2-DPO-7B

Figure 16: Confidence distributions of models on TruthfulQA before (top) and after (bottom) RLHF.

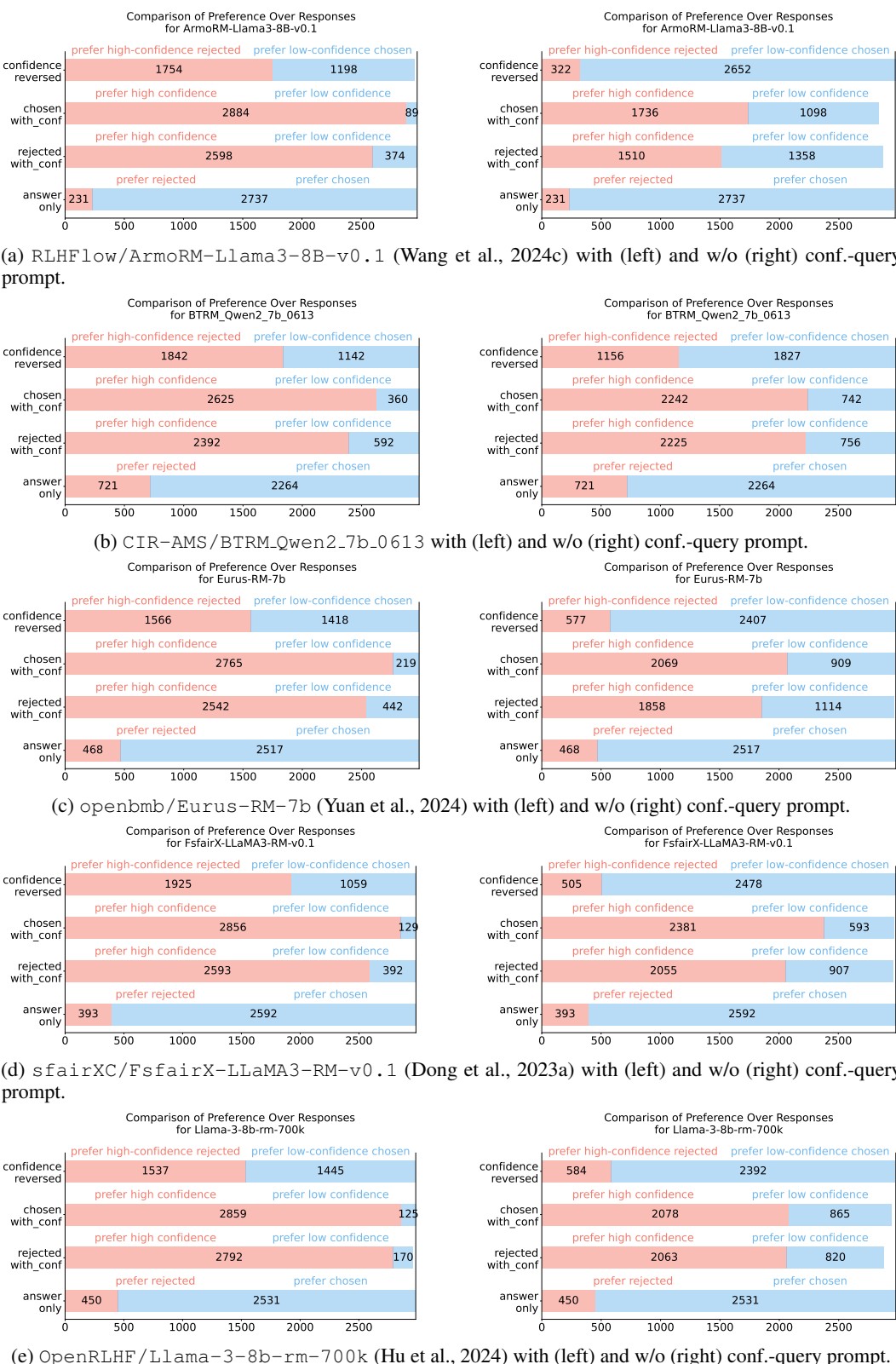

(a) `RLHFlow/ArmoRM-Llama3-8B-v0.1` (Wang et al., 2024c) with (left) and w/o (right) conf.-query prompt.

(b) `CIR-AMS/BTRM_Qwen2_7b_0613` with (left) and w/o (right) conf.-query prompt.

(c) `openbmb/Eurus-RM-7b` (Yuan et al., 2024) with (left) and w/o (right) conf.-query prompt.

(d) `sfairXC/FsfairX-LLaMA3-RM-v0.1` (Dong et al., 2023a) with (left) and w/o (right) conf.-query prompt.

(e) `OpenRLHF/Llama-3-8b-rm-700k` (Hu et al., 2024) with (left) and w/o (right) conf.-query prompt.

Figure 17: Preference Distributions for various reward models across four modes (Part 1). The left follows the same setting in preliminary experiments, while the right represents the setting where all confidence-query system prompts are removed, and only random confidence scores are appended.

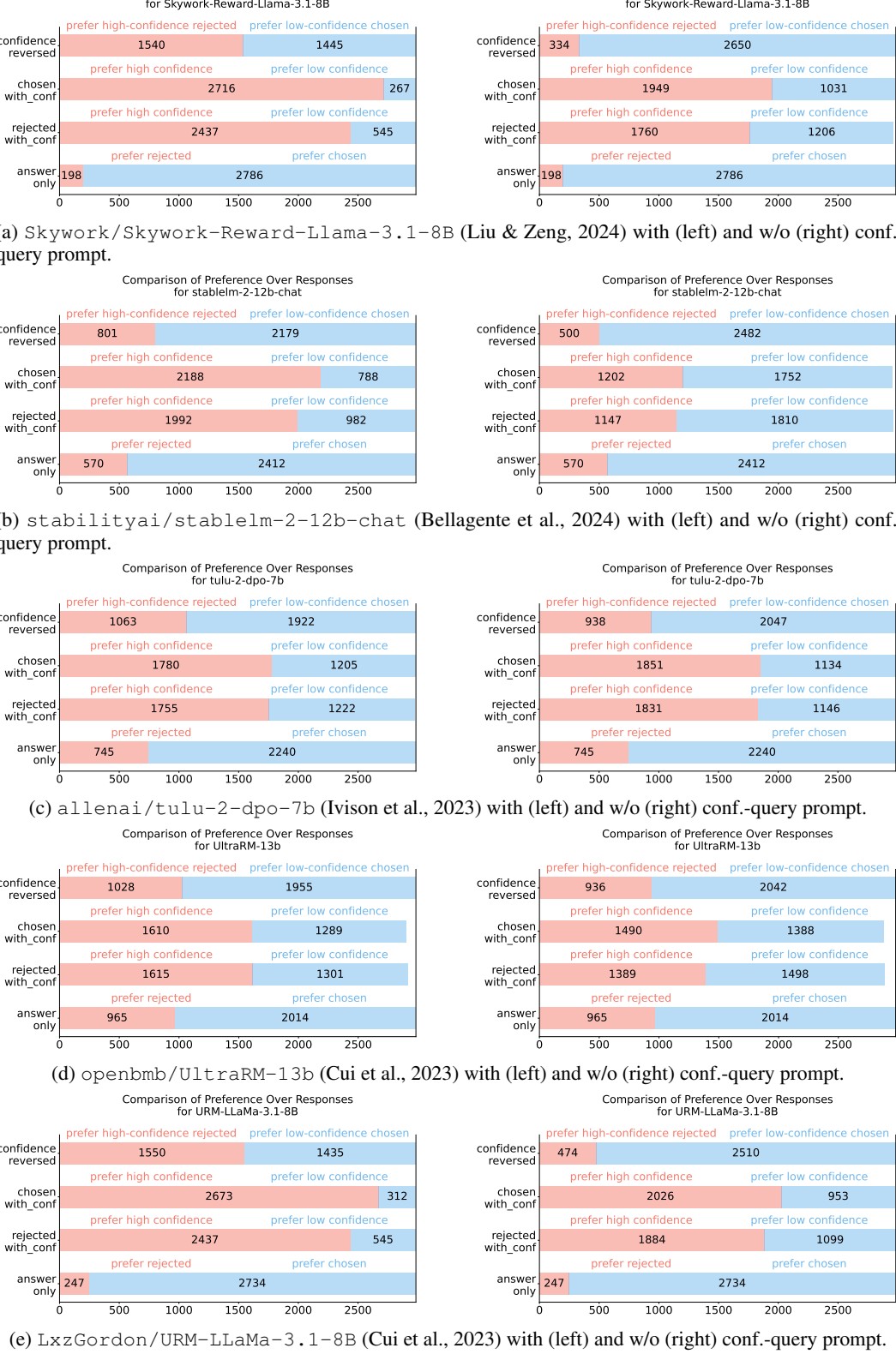

(a) `Skywork/Skywork-Reward-Llama-3.1-8B` (Liu & Zeng, 2024) with (left) and w/o (right) conf.-query prompt.

(b) `stabilityai/stablelm-2-12b-chat` (Bellagente et al., 2024) with (left) and w/o (right) conf.-query prompt.

(c) `allenai/tulu-2-dpo-7b` (Ivison et al., 2023) with (left) and w/o (right) conf.-query prompt.

(d) `openbmb/UltraRM-13b` (Cui et al., 2023) with (left) and w/o (right) conf.-query prompt.

(e) `LxzGordon/URM-LLaMa-3.1-8B` (Cui et al., 2023) with (left) and w/o (right) conf.-query prompt.

Figure 18: Preference Distributions for various reward models across four modes (Part 2). The left follows the same setting in preliminary experiments, while the right represents the setting where all confidence-query system prompts are removed, and only random confidence scores are appended.

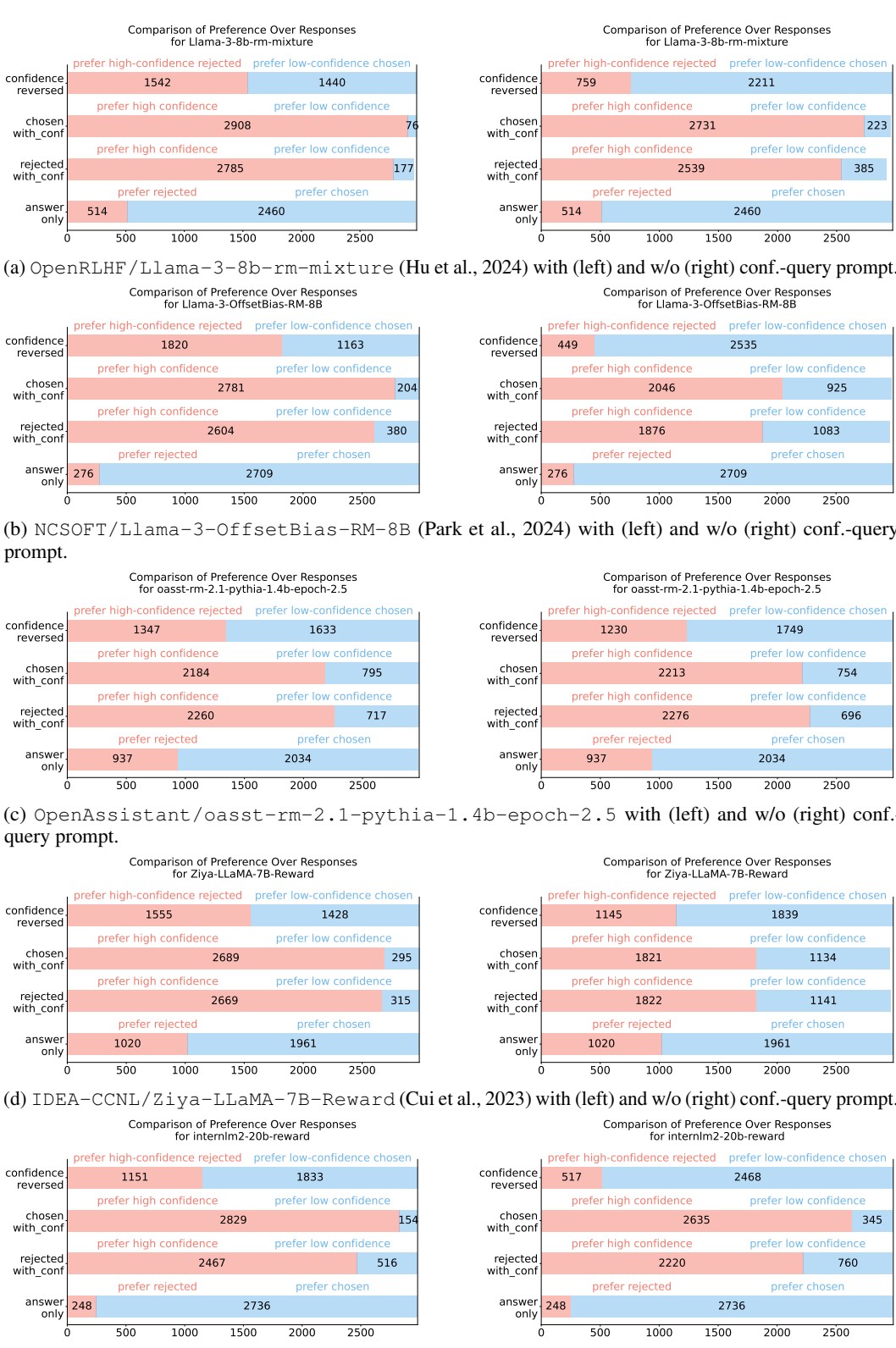

(a) `OpenRLHF/Llama-3-8b-rm-mixture` (Hu et al., 2024) with (left) and w/o (right) conf.-query prompt.

(b) `NCSOFT/Llama-3-OffsetBias-RM-8B` (Park et al., 2024) with (left) and w/o (right) conf.-query prompt.

(c) `OpenAssistant/oasst-rm-2.1-pythia-1.4b-epoch-2.5` with (left) and w/o (right) conf.-query prompt.

(d) `IDEA-CCNL/Ziya-LLaMA-7B-Reward` (Cui et al., 2023) with (left) and w/o (right) conf.-query prompt.

(e) `internlm/internlm2-20b-reward` with (left) and w/o (right) conf.-query prompt.

Figure 19: Preference Distributions for various reward models across four modes (Part 3). The left follows the same setting in preliminary experiments, while the right represents the setting where all confidence-query system prompts are removed, and only random confidence scores are appended.

A concern arises that the reward model may be influenced by the inclusion of the confidence-query system prompt, which is designed to ensure the model verbalizes its confidence level. To investigate the impact of this system prompt, we conduct additional experiments with and without its inclusion.

As shown in Figure 17, 18, and 19, the plots on the left follow the configuration outlined in preliminary experiments, where a confidence-query system prompt is prepended and random confidence scores are appended to model responses. These plots clearly demonstrate that all tested reward models exhibit a biased preference towards high-confidence responses, with the degree of bias varying across models. On the right, we evaluate four modes, but this time *without the confidence-query system prompts*, and only random confidence scores are appended to the model responses. For example, in REJECTED_WITH_CONF, the comparison involves the same chosen responses with a high confidence score versus a low confidence score. The results reveal a similar phenomenon, although the bias is more subtle in this setting, indicating the potential influence of the confidence-query system prompt.

### E.3 CALIBRATED REWARD MODELS

Section 4.2 highlights the preference distributions of our calibrated reward model compared to the pre-calibrated version for Llama3-8B on REJECTED_WITH_CONF. In this section, we present the complete set of results and extend the analysis to include the Mistral-7B model, providing a more comprehensive evaluation of the calibrated reward models' performance across different architectures.

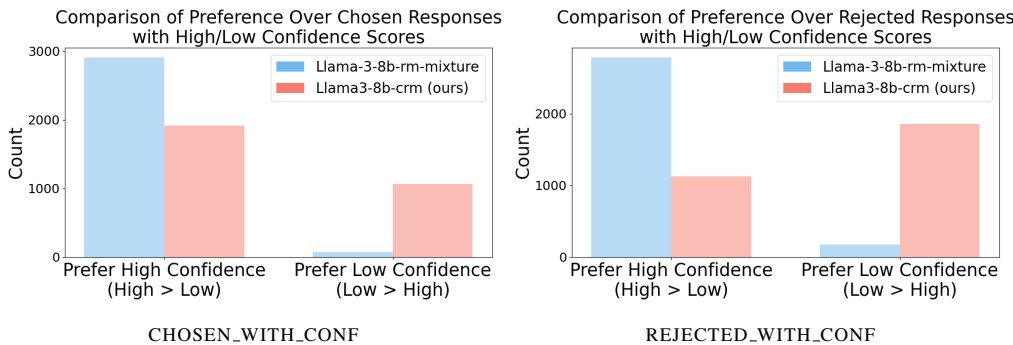

Figure 20: Comparison of preference distributions between the calibrated reward model Llama-3-8b-crm and the pre-calibrated version Llama-3-8b-rm-mixture on two modes.

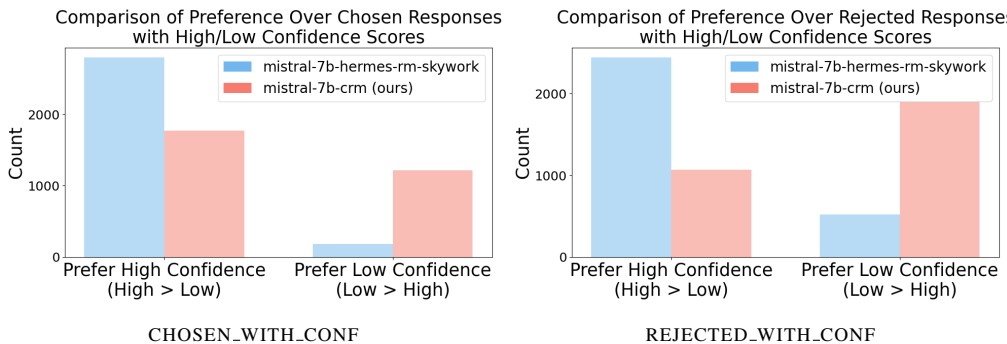

Figure 21: Comparison of preference distributions between the calibrated reward model Mistral-7B-crm and the pre-calibrated version Mistral-7B-RM on two modes.

As shown in Figure 20 and 21, both calibrated models exhibit a similar trend. When evaluated on chosen responses with high and low confidence scores, the calibrated reward models are less certain than their pre-calibrated counterparts. Additionally, when evaluated on rejected responses with high and low confidence scores, both calibrated models show a preference for low-confidence responses, indicating improved capability of our calibrated models in identifying overconfident model responses.

### E.4 VISUALIZATION OF THE CONFIDENCE DISTRIBUTION

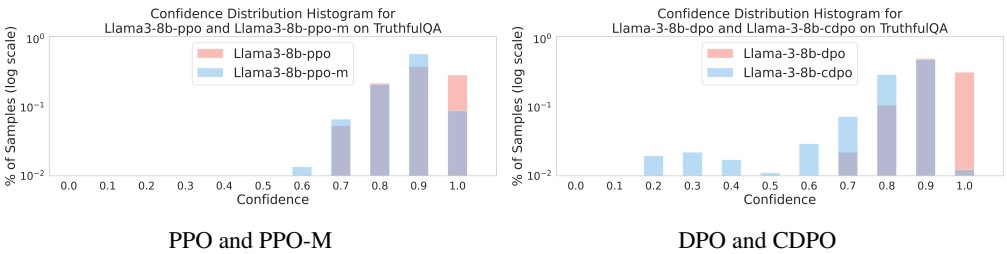

Figure 22: Confidence distributions of PPO and PPO-M (left) and DPO and CDPO (right).

In Figure 22, we present the confidence distributions of the PPO and PPO-M models on the left, and the DPO and CDPO models on the right. Notably, the confidence distribution for PPO-M is slightly shifted to the left relative to PPO, indicating a reduction in high-confidence scores (e.g., confidence level 10, representing a highly overconfident state) and an increase in lower-confidence categories. For CDPO, this phenomenon is even more pronounced; compared to DPO, the confidence distribution of CDPO is more dispersed across categories, with a noticeable increase in lower-confidence levels.

### E.5 MODEL LOGITS FOR CONFIDENCE SCORES

Figure 23 presents the density distribution of numbers 0 to 10 based on the log probabilities extracted from model responses on the TruthfulQA dataset for both PPO and PPO-M models. Specifically, we forward the model responses and examine the log probabilities at the position corresponding to the original confidence score within the response. We then analyze the log probabilities of other numbers at the same position. The figure reveals that certain numbers exhibit notably high density. For instance, the PPO model exhibits a high density for the number 10, while the PPO-M model favors the number 9. This non-uniform distribution of log probabilities indicates that the model does not generate numbers randomly at the confidence score position but instead favors specific numbers.

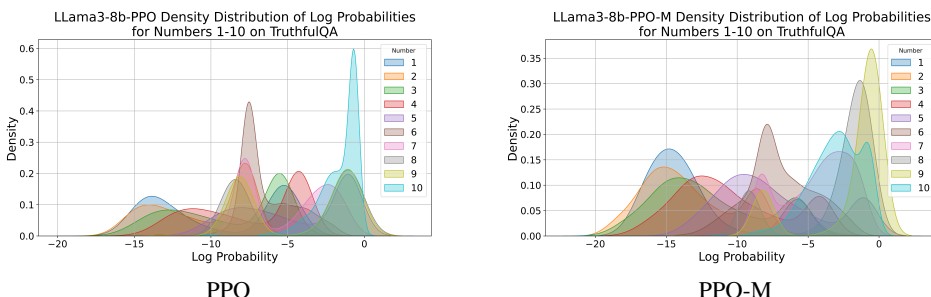

Figure 23: Density Plot of LogProb for Confidence Scores for PPO and PPO-M on TruthfulQA.

### E.6 PARAMETER SENSITIVITY

In Eq. 3, we introduce a reward adjustment factor $\gamma$, defined as $\gamma = w * (\hat{r}_i - \Delta r_t) * (s_i - 0.5)$. Here $w$ represents a scaling coefficient set to 2.0 in our main results. To evaluate the impact of $w$, we conduct a hyperparameter sensitivity study, detailed in this section. The results, presented in Table 10, reveal a clear positive correlation between calibration performance and $w$, and a negative correlation between model instruction-following performance and $w$. This demonstrates a trade-off between calibration effectiveness and model instruction-following capabilities as $w$ increases. Increasing $w$ from 0.5 to 2.0 significantly enhances calibration performance, as indicated by a decrease in ECE. However, this improvement is accompanied by a slight reduction in MT-Bench and Arena-Hard scores. Based on our primary focus on confidence calibration, we select $w = 2.0$ for the main results.

| $w$ | MT/Arena-Hard | GSM8K | | | SciQ | | | CommonsenseQA | | |
|---|---|---|---|---|---|---|---|---|---|---|
| | | ECE ↓ | AUC ↑ | ACC ↑ | ECE ↓ | AUC ↑ | ACC ↑ | ECE ↓ | AUC ↑ | ACC ↑ |
| 0.5 | 8.03 / 14.7 | 0.8792 | 0.521 | 0.1099 | 0.0703 | 0.6031 | 0.896 | 0.1552 | 0.5678 | 0.7674 |
| 1.0 | 7.91 / 13.8 | 0.8238 | 0.4937 | 0.119 | 0.0087 | 0.578 | 0.898 | 0.1153 | 0.585 | 0.7625 |
| 2.0 | 7.87 / 13.7 | 0.8025 | 0.5342 | 0.1046 | 0.0319 | 0.5892 | 0.906 | 0.0457 | 0.5835 | 0.7699 |
| $w$ | MT/Arena-Hard | TruthfulQA | | | Object Counting | | | Professional Knowledge | | |
| | | ECE ↓ | AUC ↑ | ACC ↑ | ECE ↓ | AUC ↑ | ACC ↑ | ECE ↓ | AUC ↑ | ACC ↑ |
| 0.5 | 8.03 / 14.7 | 0.4428 | 0.5549 | 0.4553 | 0.4856 | 0.5036 | 0.512 | 0.4286 | 0.5027 | 0.4906 |
| 1.0 | 7.91 / 13.8 | 0.4104 | 0.515 | 0.4492 | 0.4774 | 0.5118 | 0.496 | 0.383 | 0.509 | 0.4902 |
| 2.0 | 7.87 / 13.7 | 0.3486 | 0.4856 | 0.4455 | 0.4405 | 0.5309 | 0.509 | 0.3318 | 0.5263 | 0.4798 |

Table 10: Performance of PPO-C with different $w$ coefficient on `Llama3-8B`. Prompts: DA.

| $\alpha$ | MT-Bench | GSM8K | | | SciQ | | | CommonsenseQA | | |
|---|---|---|---|---|---|---|---|---|---|---|
| | | ECE ↓ | AUC ↑ | ACC ↑ | ECE ↓ | AUC ↑ | ACC ↑ | ECE ↓ | AUC ↑ | ACC ↑ |
| 0 | 7.97 | 0.8832 | 0.5 | 0.1168 | 0.0967 | 0.5244 | 0.902 | 0.2251 | 0.5111 | 0.7715 |
| 0.1 | 7.87 | 0.8025 | 0.5343 | 0.1046 | 0.0319 | 0.5892 | 0.906 | 0.0457 | 0.5835 | 0.7699 |
| 1.0 | 7.97 | 0.8658 | 0.5009 | 0.1114 | 0.0373 | 0.6426 | 0.905 | 0.0821 | 0.5646 | 0.7756 |
| $\alpha$ | MT-Bench | TruthfulQA | | | Object Counting | | | Professional Knowledge | | |
| | | ECE ↓ | AUC ↑ | ACC ↑ | ECE ↓ | AUC ↑ | ACC ↑ | ECE ↓ | AUC ↑ | ACC ↑ |
| 0 | 7.97 | 0.5502 | 0.5332 | 0.437 | 0.4947 | 0.501 | 0.505 | 0.4877 | 0.4985 | 0.5072 |
| 0.1 | 7.87 | 0.3486 | 0.4856 | 0.4455 | 0.4405 | 0.5309 | 0.509 | 0.3318 | 0.5263 | 0.4798 |
| 1.0 | 7.97 | 0.3846 | 0.524 | 0.4443 | 0.4899 | 0.4985 | 0.506 | 0.381 | 0.52 | 0.4728 |

Table 11: Difference-Based PPO-C with different $\alpha$ for $\Delta r$ on `Llama3-8B`. Prompts: DA.

| $\alpha$ | MT-Bench | GSM8K | | | SciQ | | | CommonsenseQA | | |
|---|---|---|---|---|---|---|---|---|---|---|
| | | ECE ↓ | AUC ↑ | ACC ↑ | ECE ↓ | AUC ↑ | ACC ↑ | ECE ↓ | AUC ↑ | ACC ↑ |
| 0 | 7.79 | 0.8833 | 0.5034 | 0.116 | 0.1056 | 0.5238 | 0.891 | 0.2178 | 0.5568 | 0.7649 |
| 0.1 | 8.05 | 0.8638 | 0.516 | 0.1031 | 0.0282 | 0.6513 | 0.904 | 0.1286 | 0.5621 | 0.7756 |
| 1.0 | 8.03 | 0.8827 | 0.5112 | 0.1145 | 0.0849 | 0.5493 | 0.907 | 0.1992 | 0.5632 | 0.7625 |
| $\alpha$ | MT-Bench | TruthfulQA | | | Object Counting | | | Professional Knowledge | | |
| | | ECE ↓ | AUC ↑ | ACC ↑ | ECE ↓ | AUC ↑ | ACC ↑ | ECE ↓ | AUC ↑ | ACC ↑ |
| 0 | 7.79 | 0.5185 | 0.5655 | 0.4394 | 0.4948 | 0.498 | 0.505 | 0.4753 | 0.5119 | 0.5024 |
| 0.1 | 8.05 | 0.4426 | 0.5303 | 0.4431 | 0.4839 | 0.5178 | 0.503 | 0.3949 | 0.4902 | 0.502 |
| 1.0 | 8.03 | 0.4965 | 0.5595 | 0.4333 | 0.4797 | 0.5011 | 0.52 | 0.4614 | 0.4968 | 0.4935 |

Table 12: Threshold-Based PPO-C with different $\alpha$ for $\Delta r$ on `Llama3-8B`. Prompts: DA.

| Percentage | MT-Bench | GSM8K | | | SciQ | | | CommonsenseQA | | |
|---|---|---|---|---|---|---|---|---|---|---|
| | | ECE ↓ | AUC ↑ | ACC ↑ | ECE ↓ | AUC ↑ | ACC ↑ | ECE ↓ | AUC ↑ | ACC ↑ |
| 0.25 | 8.05 | 0.8393 | 0.57 | 0.119 | 0.0267 | 0.6115 | 0.898 | 0.1206 | 0.5568 | 0.7707 |
| 0.5 | 7.88 | 0.86 | 0.5185 | 0.1031 | 0.0389 | 0.5829 | 0.896 | 0.134 | 0.5399 | 0.7682 |
| 1.0 | 7.74 | 0.8608 | 0.5065 | 0.1243 | 0.0471 | 0.7165 | 0.898 | 0.074 | 0.6341 | 0.7658 |
| Percentage | MT-Bench | TruthfulQA | | | Object Counting | | | Professional Knowledge | | |
| | | ECE ↓ | AUC ↑ | ACC ↑ | ECE ↓ | AUC ↑ | ACC ↑ | ECE ↓ | AUC ↑ | ACC ↑ |
| 0.25 | 8.05 | 0.3991 | 0.5813 | 0.47 | 0.4789 | 0.5227 | 0.505 | 0.3848 | 0.4926 | 0.502 |
| 0.5 | 7.88 | 0.4453 | 0.5283 | 0.4357 | 0.5119 | 0.5413 | 0.473 | 0.3988 | 0.5221 | 0.4935 |
| 1.0 | 7.74 | 0.3438 | 0.5737 | 0.4786 | 0.5087 | 0.5052 | 0.487 | 0.3501 | 0.5184 | 0.502 |

Table 13: Performance of PPO-M on downstream tasks using Prompt Dataset with various percentage of single-turn prompts prepending confidence-query system prompts on `Llama3-8B`. Prompts: DA.

Tables 11 and 12 present ablation studies on $\alpha$, the decay factor for the exponential average, for both difference-based and threshold-based PPO-C. This parameter controls how quickly the exponential average adapts to new data and reflects recent model performance. For the main results, we set $\alpha = 0.1$, a commonly used value for exponential averages, as it balances stability with filtering out short-term variability. We compare this to $\alpha = 1.0$, where the exponential average is updated to match the batch mean at each iteration, and $\alpha = 0.0$, where it remains fixed at its initial value (in this case, it is initialized as the reward mean on the evaluation set when the reward model is trained). As shown in the tables, $\alpha = 1.0$ leads to a notable overll decline in calibration performance and a slight increase in the MT-Bench score for difference-based PPO-C. Similarly, $\alpha = 0.0$ results in consistently inferior performance compared to $\alpha = 0.1$ in both calibration and MT-Bench scores.

### E.7 IMPACT OF CONFIDENCE-QUERY SYSTEM PROMPTS

For the main experiments, we select 25% of the single-turn prompts to prepend a confidence-query system prompt. Here, we present our study on the effect of varying the percentage of single-turn prompts with this system prompt. As shown in Table 13, the impact on calibration does not show a consistent trend; however, we observe a decrease in MT-Bench scores as the percentage increases. Given our primary goal to maintain model capability while improving calibration, we opt for 25%.

### E.8 IMPACT OF COMBINING EQ. 1 AND 2

Given that Eq. 2 does not inherently enforce the preference for chosen responses over rejected ones. In this section, we compare models trained using the combined loss from Eq.1 and Eq.2 against those trained solely with Eq.2. It is important to note that we are not training the reward model from scratch; instead, we fine-tune it using the calibration dataset. As shown in Figure 14, the model trained exclusively with Eq 2 exhibits a similar ability to distinguish between chosen and rejected responses as the model trained with the combined loss. Furthermore, Table 14 shows that PPO-M, when using the reward model trained with the combined loss does not yield better calibration results.

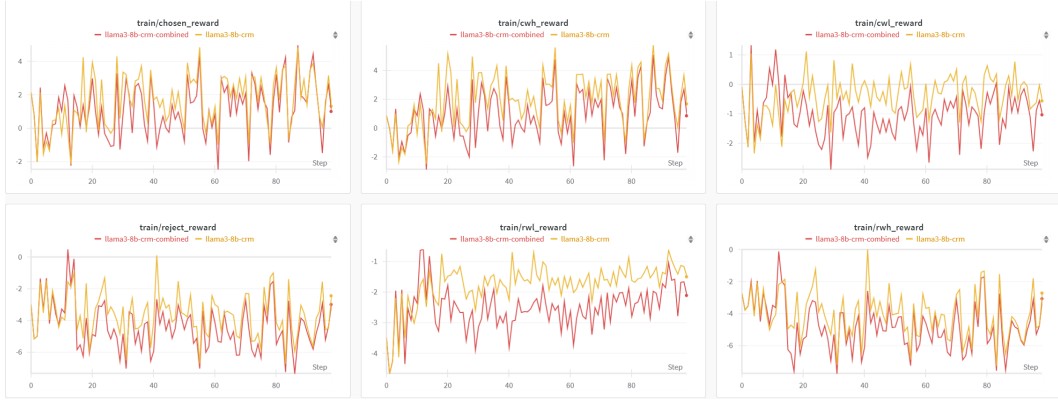

Figure 24: Training Details of reward model with Eq. 2 alone (orange) and in combination with Eq. 1 (red). Left column: reward of chosen / rejected responses. Middle column: reward of chosen responses with high confidence / reward of rejected responses with low confidence. Right column: reward of chosen responses with low confidence / reward of rejected responses with high confidence.

### E.9 COMPARING THRESHOLD-BASED VS. REWARD-AVERAGE DIFFERENCE APPROACHES

While PPO-C has demonstrated effectiveness, as shown in Table 1, it is important to explore alternative methods for adjusting reward scores to provide a broader perspective and facilitate comprehensive comparisons. In this section, we introduce a threshold-based variant of PPO-C for evaluation. Specifically, we use the reward exponential average as a threshold and employ the absolute value of the reward as a scaling factor for adjustment. The final reward in this approach is then calculated as:

$$r_i = \begin{cases} \hat{r}_i + \gamma & \text{if } \hat{r}_i \geq \Delta r_t \\ \hat{r}_i - \gamma & \text{if } \hat{r}_i < \Delta r_t \end{cases} \tag{5}$$

| Loss | MT-Bench | GSM8K | | | SciQ | | | CommonsenseQA | | |
|---|---|---|---|---|---|---|---|---|---|---|
| | | ECE ↓ | AUC ↑ | ACC ↑ | ECE ↓ | AUC ↑ | ACC ↑ | ECE ↓ | AUC ↑ | ACC ↑ |
| Eq. 2 | 8.05 | 0.8638 | 0.516 | 0.1031 | 0.0282 | 0.6513 | 0.904 | 0.1286 | 0.5621 | 0.7756 |
| Eq. 1+2 | 7.75 | 0.8891 | 0.4974 | 0.1107 | 0.1043 | 0.5186 | 0.894 | 0.2286 | 0.528 | 0.7584 |
| **Loss** | **MT-Bench** | **TruthfulQA** | | | **Object Counting** | | | **Professional Knowledge** | | |
| | | ECE ↓ | AUC ↑ | ACC ↑ | ECE ↓ | AUC ↑ | ACC ↑ | ECE ↓ | AUC ↑ | ACC ↑ |
| Eq. 2 | 8.05 | 0.4426 | 0.5303 | 0.4431 | 0.4839 | 0.5178 | 0.503 | 0.3949 | 0.4902 | 0.502 |
| Eq. 1+2 | 7.75 | 0.5006 | 0.564 | 0.4565 | 0.518 | 0.5 | 0.482 | 0.4786 | 0.4964 | 0.5061 |

Table 14: PPO-M with the reward model trained using two losses on `Llama3-8B`. Prompts: DA.

where $\gamma = w * |\hat{r}_i| * (s_i - 0.5)$. As shown in Table 15, we refer to this new threshold-based PPO-C variant as *Threshold* and the original PPO-C as *Difference* in the table. The threshold-based PPO-C demonstrates promising results across six datasets. It also exhibits a similar trade-off trand between calibration and model instruction-following capabilities as $w$ increases. These results suggest that threshold-based approach may serve as a viable alternative for calibrating reward scores during PPO.

| Method | w | MT | GSM8K | | | SciQ | | | CommonsenseQA | | |
|---|---|---|---|---|---|---|---|---|---|---|---|
| | | | ECE ↓ | AUC ↑ | ACC ↑ | ECE ↓ | AUC ↑ | ACC ↑ | ECE ↓ | AUC ↑ | ACC ↑ |
| Threshold | 0.5 | 8.05 | 0.8638 | 0.516 | 0.1031 | 0.0282 | 0.6513 | 0.904 | 0.1286 | 0.5621 | 0.7756 |
| Threshold | 1.0 | 7.76 | 0.8261 | 0.501 | 0.1092 | 0.0075 | 0.5641 | 0.903 | 0.1025 | 0.5076 | 0.7805 |
| Difference | 0.5 | 8.03 | 0.8792 | 0.521 | 0.1099 | 0.0703 | 0.6031 | 0.896 | 0.1552 | 0.5678 | 0.7674 |
| Difference | 1.0 | 7.91 | 0.8238 | 0.4937 | 0.119 | 0.0087 | 0.578 | 0.898 | 0.1153 | 0.585 | 0.7625 |
| **Method** | **w** | **MT** | **TruthfulQA** | | | **Object Counting** | | | **Professional Knowledge** | | |
| | | | ECE ↓ | AUC ↑ | ACC ↑ | ECE ↓ | AUC ↑ | ACC ↑ | ECE ↓ | AUC ↑ | ACC ↑ |
| Threshold | 0.5 | 8.05 | 0.4426 | 0.5303 | 0.4431 | 0.4839 | 0.5178 | 0.503 | 0.3949 | 0.4902 | 0.502 |
| Threshold | 1.0 | 7.76 | 0.4271 | 0.5207 | 0.4345 | 0.4709 | 0.5318 | 0.505 | 0.388 | 0.5069 | 0.4883 |
| Difference | 0.5 | 8.03 | 0.4428 | 0.5549 | 0.4553 | 0.4856 | 0.5036 | 0.512 | 0.4286 | 0.5027 | 0.4906 |
| Difference | 1.0 | 7.91 | 0.4104 | 0.515 | 0.4492 | 0.4774 | 0.5118 | 0.496 | 0.383 | 0.509 | 0.4902 |

Table 15: Comparison of Threshold-Based and Diff-Based PPO-C on `Llama3-8B`. Prompts: DA.

### E.10 CAN PPO-M AND PPO-C BE COMBINED?

| | MT-Bench | Arena-Hard | GSM8K | | | SciQ | | | CommonsenseQA | | |
|---|---|---|---|---|---|---|---|---|---|---|---|
| | | | ECE ↓ | AUC ↑ | ACC ↑ | ECE ↓ | AUC ↑ | ACC ↑ | ECE ↓ | AUC ↑ | ACC ↑ |
| DA | 7.82 | 14.7 | 0.8774 | 0.6199 | 0.0538 | 0.104 | 0.5834 | 0.879 | 0.1774 | 0.5837 | 0.7617 |
| CoT | 7.82 | 14.7 | 0.2123 | 0.5317 | 0.7794 | 0.0909 | 0.6641 | 0.884 | 0.1957 | 0.6335 | 0.7297 |
| | **MT-Bench** | **Arena-Hard** | **TruthfulQA** | | | **Object Counting** | | | **Professional Knowledge** | | |
| | | | ECE ↓ | AUC ↑ | ACC ↑ | ECE ↓ | AUC ↑ | ACC ↑ | ECE ↓ | AUC ↑ | ACC ↑ |
| DA | 7.82 | 14.7 | 0.4654 | 0.5178 | 0.4345 | 0.4927 | 0.5 | 0.507 | 0.5005 | 0.5287 | 0.4216 |
| CoT | 7.82 | 14.7 | 0.4561 | 0.5656 | 0.4419 | 0.2843 | 0.5 | 0.715 | 0.4525 | 0.5793 | 0.4439 |

Table 16: Performance of PPO-Combine on `Llama3-8B` across six datasets.

Since PPO-M and PPO-C operate independently, this section explores the potential of combining these methods. Specifically, the calibrated reward models using Eq. 2 are employed in conjunction with the calibrated reward calculation from PPO-C to generate reward scores. The results, presented in Table 16, indicate that the combined approach does not outperform the individual methods and, in

| Methods | | GSM8K | | | SciQ | | | CommonsenseQA | | |
|---|---|---|---|---|---|---|---|---|---|---|
| | | ECE ↓ | AUC ↑ | ACC ↑ | ECE ↓ | AUC ↑ | ACC ↑ | ECE ↓ | AUC ↑ | ACC ↑ |
| DA | SFT | 0.8783 | 0.5292 | 0.0773 | 0.1681 | 0.5253 | 0.801 | 0.3913 | 0.5294 | 0.5528 |
| | DPO | 0.904 | 0.5381 | 0.0834 | 0.1085 | 0.561 | 0.886 | 0.3011 | 0.535 | 0.6871 |
| | DPO† | 0.8861 | 0.5203 | 0.097 | 0.1103 | 0.5626 | **0.881** | 0.3004 | 0.5409 | 0.683 |
| | CDPO | **0.5664** | **0.5389** | **0.1024** | **0.0143** | **0.6497** | 0.877 | **0.1697** | **0.5815** | **0.6912** |
| CoT | SFT | 0.6473 | 0.5508 | 0.326 | 0.1699 | 0.5816 | 0.803 | 0.3293 | 0.588 | 0.579 |
| | DPO | 0.4159 | 0.5452 | 0.577 | 0.113 | 0.6376 | 0.858 | 0.2621 | 0.6295 | 0.6593 |
| | DPO† | 0.452 | 0.5456 | **0.539** | 0.0964 | 0.6614 | **0.876** | 0.235 | 0.5973 | 0.6749 |
| | CDPO | **0.3313** | **0.6054** | 0.5277 | **0.0386** | **0.7036** | 0.86 | **0.1269** | **0.6685** | 0.6798 |

| Methods | | TruthfulQA | | | Object Counting | | | Professional Knowledge | | |
|---|---|---|---|---|---|---|---|---|---|---|
| | | ECE ↓ | AUC ↑ | ACC ↑ | ECE ↓ | AUC ↑ | ACC ↑ | ECE ↓ | AUC ↑ | ACC ↑ |
| DA | SFT | 0.592 | 0.5388 | 0.3256 | 0.5964 | 0.4938 | 0.395 | 0.5109 | 0.5189 | 0.4127 |
| | DPO | 0.6126 | 0.5581 | 0.3525 | 0.5848 | 0.4996 | 0.415 | 0.4764 | 0.4992 | 0.495 |
| | DPO† | 0.5647 | 0.5886 | 0.3856 | 0.5999 | 0.5008 | 0.4 | 0.467 | 0.5153 | **0.4939** |
| | CDPO | **0.4022** | **0.6194** | **0.3929** | **0.4662** | **0.5262** | **0.422** | **0.3525** | **0.5581** | 0.4898 |
| CoT | SFT | 0.5259 | 0.5698 | 0.3782 | 0.5388 | 0.5126 | 0.45 | 0.5091 | 0.5457 | 0.4068 |
| | DPO | 0.5188 | 0.5822 | 0.4088 | 0.3520 | 0.5000 | 0.6480 | 0.4289 | 0.5700 | 0.4831 |
| | DPO† | 0.4931 | 0.6111 | 0.4113 | 0.3783 | 0.5018 | **0.621** | 0.4312 | 0.562 | **0.4694** |
| | CDPO | **0.3651** | **0.634** | **0.4345** | **0.3488** | **0.5286** | 0.567 | **0.3349** | **0.6303** | 0.4609 |

Table 18: Performance comparison of SFT, DPO, DPO†, and CDPO across six datasets using `Llama3-8B`. SFT and DPO denote the reference and trained DPO models, respectively. DPO† and CDPO initiate from the trained DPO checkpoint; DPO† applies standard DPO on the calibration dataset, focusing on chosen and rejected pairs to assess the impact of training with additional data.

some cases, leads to a decline in performance. We hypothesize that this outcome arises because the calibrated reward model is trained specifically on responses incorporating confidence scores, which are optimized to produce unbiased rewards. Consequently, removing these confidence scores to estimate rewards based on their difference from exponential average dynamic may be inappropriate.

### E.11 EXTENSION TO DPO

In Section 5.2, we present the results of extending PPO-M to DPO training on `Mistral-7B`. In this section, we include additional results for `Llama3-8B`. As shown in Table 18 and 17, CDPO effectively reduces ECE and increases AUC, mirroring the trend observed with `Mistral-7B`, while maintaining performance on MT-Bench. However, we observe a slight performance degradation on Arena-Hard using either DPO† or CDPO. This issue may arise from insufficient hyperparameter tuning or inherent limitations in the structure of the calibration dataset, which we leave for future research.

| Model | Method | MT-Bench ↑ | Arena-Hard ↑ |
|---|---|---|---|
| **Llama3-8B** | SFT | 6.44 (6.6) | 3.1 (3.3) |
| | DPO | 7.67 (7.7) | 15.9 (15.9) |
| | DPO† | 7.52 | **15.2** |
| | CDPO | **7.68** | 14.7 |

Table 17: Comparison of DPO and CDPO on MT-Bench And Arena-Hard for `Llama3-8B`. Numbers in parenthesis are from Meng et al. (2024).

