# OpenReview forum: "Taming Overconfidence in LLMs: Reward Calibration in RLHF"
_ICLR.cc/2025/Conference — ICLR 2025 Poster_

### Official Review · Reviewer_BbUH · 2024-10-31

**Soundness:** 2
**Presentation:** 4
**Contribution:** 2
**Rating:** 5
**Confidence:** 3

**Summary:**

This paper identifies a systematic bias in reward models that favors high-confidence responses regardless of their actual quality. They propose two solutions for this bias issue: PPO-M, which calibrates reward modeling by aligning confidence levels with response quality, and PPO-C, which adjusts standard reward model scores during PPO training. Empirical results show their methods can reduce expected calibration error without compromising accuracy or instruction-following ability in open-ended generation.

**Strengths:**

1. Good motivation to identify the systematic bias and propose intuitive solutions.
2. The presentation is clear and easy to follow, with a logical and well-reasoned train of thought.
3. Sufficient empirical results to justify the effectiveness of proposed methods

**Weaknesses:**

1. Regarding the proposed PPO-M, there is no guarantee that the model would prefer $(y_c,h_c)$ over $(y_r,l_r)$. This could be a potential reason why the calibrated model results in lower ACC metrics than the uncalibrated PPO shown in the tables. Although the systematic bias may cause the model to prefer $h_c$ over $l_r$, PPO-M lacks an explicit mechanism to enforce this preference. Have you considered jointly optimizing (1) and (2) to enforce $y_c$ over $y_r$ as well?

2. Since PPO-C depends on a running average reward for confidence calibration, the $\alpha$ affecting the moving average could be an important factor. However, there seems no evaluation of how $\alpha$ affects PPO-C. I suggest experiments to show how varying $\alpha$ influences reward magnitude and performance. Moreover, are you assuming that $\hat{r}(y_c)>\Delta r$ and $\hat{r}(y_r)<\Delta r$? If so, would be helpful to explicitly state the assumption and analyze its appropriateness.

3. The design and evaluation of the proposed algorithms, as noted above, lack clarity or rigor. The formulations would benefit from a more detailed and persuasive explanation. Ideally, theoretical support should be provided to justify that the approach achieves debiasing without compromising model performance. At present, the methods appear to rely heavily on heuristics rather than principled algorithm design, which diminishes the significance of the contribution.

**Questions:**

1. Why not add something similar to (1) onto (2) to enforce the preference of $(y_c,h_c)$ over $(y_r,l_r)$?
2. Why do PPO-M and PPO-C often result in lower ACC?
3. How does $\alpha$ affect the results of PPO-C?
4. Are you assuming that $\hat{r}(y_c)>\Delta r$ and $\hat{r}(y_r)<\Delta r$?

Overall, It is a well-motivated work presented in a structured manner. However, the contributions lack substantial impact. The proposed algorithms resemble engineering techniques rather than rigorous, principled approaches.

## Lower confidence after discussion
I maintain my critique of the heuristic algorithm's lack of rigorous design but am less certain if this empirical research meets ICLR standards. I've reduced my confidence to 3.

To be honest, the algorithm's design and naming seem somewhat inelegant and naive. While empirical results are solid, nowadays it has become increasingly difficult to determine whether their success stems from effective code-level implementation, a limited dataset, or a genuinely strong algorithm.

Since other reviewers have given higher ratings, I am open to its acceptance as a poster. That said, I would be deeply disappointed if it were elevated to the level of a spotlight or oral presentation, as I believe this would set a poor precedent for ICLR's standards.

**Details Of Ethics Concerns:**

There are no apparent violations of the code; however, it is worth noting that this paper is available on Arxiv.

---

> ### Author Response · Authors · 2024-11-19
> **Response to Reviewer BbUH (Part 1)**
>
> Thank you for reviewing our paper and for your valuable feedback. Below, we address your concerns point by point. We would appreciate it if you could let us know whether your concerns are addressed by our response.
>
> **Q1:** Have you considered jointly optimizing (1) and (2) to enforce $y_c$ over $y_r$ as well.
>
> **A1:** Thank you for your valuable insight. Equation (2) is intended to function as a stand-alone objective. In our experiments, when **fine-tuning an existing reward model** on our calibration dataset, we did not observe any diminishment in the model’s ability to distinguish between chosen and rejected responses. As shown in Figure 6 of Section 4.2 in the updated PDF, we provide a direct comparison between our calibrated reward model and its pre-calibrated version. Specifically, in the answer_only row, which uses the original RewardBench datasets without modifications, **the chosen/rejected preference ratio remains largely unchanged**.
>
> Additionally, in Appendix E.8 and Figure 24, we present the training plots of the reward model trained using equations (1) and (2) in combination, compared to training with equation (2) alone. The model trained solely with equation (2) maintains a similar trend of distinguishing between chosen and rejected responses as the model trained with the combined loss.
>
> We also conducted experiments comparing PPO-M using the reward model trained with equation (2) alone versus in combination with equation (1), as shown in Table 13 of the updated manuscript. For your convenience, we have provided the table below.
>
> ---
> | Loss                             | MT-Bench | ECE ↓ (GSM8K) | AUC ↑ (GSM8K) | ACC ↑ (GSM8K) | ECE ↓ (SciQ) | AUC ↑ (SciQ) | ACC ↑ (SciQ) | ECE ↓ (CommonsenseQA) | AUC ↑ (CommonsenseQA) | ACC ↑ (CommonsenseQA) |
> |----------------------------------|----------|---------------|---------------|--------------|--------------|--------------|--------------|------------------------|-----------------------|-----------------------|
> | (2)          | 8.05     | 0.8638        | 0.516         | 0.1031       | 0.0282       | 0.6513       | 0.904        | 0.1286                 | 0.5621                | 0.7756                |
> | (1) + (2) | 7.75     | 0.8891        | 0.4974        | 0.1107       | 0.1043       | 0.5186       | 0.894        | 0.2286                 | 0.528                 | 0.7584                |
>
> ---
>
> | Loss                             | MT-Bench | ECE ↓ (TruthfulQA) | AUC ↑ (TruthfulQA) | ACC ↑ (TruthfulQA) | ECE ↓ (Object Counting) | AUC ↑ (Object Counting) | ACC ↑ (Object Counting) | ECE ↓ (Professional Knowledge) | AUC ↑ (Professional Knowledge) | ACC ↑ (Professional Knowledge) |
> |----------------------------------|----------|---------------------|--------------------|-------------------|-------------------------|-------------------------|-------------------------|-------------------------------|-------------------------------|-------------------------------|
> | (1)           | 8.05     | 0.4426              | 0.5303             | 0.4431            | 0.4839                  | 0.5178                  | 0.503                  | 0.3949                        | 0.4902                        | 0.502                         |
> | (1) + (2) | 7.75     | 0.5006              | 0.564              | 0.4565            | 0.518                   | 0.5                     | 0.482                  | 0.4786                        | 0.4964                        | 0.5061                        |
> ---
>
> To summarize:
> * The results indicate that PPO-M using the reward model trained with the combined loss does not achieve better calibration results. Moreover, combining equations (1) and (2) would require additional computational resources since we need to process two extra samples. Given these observations, we have chosen to set equation (2) as a standalone loss.
>
> Furthermore, in Figure 24 (middle column), when comparing cwh_reward (the reward for a chosen response with high confidence, $(y_c, h_c)$ and rwl_reward (the reward for a rejected response with low confidence, $(y_r, l_r)$, we observe that the rewards of $(y_c, h_c)$ consistently exceed the rewards of $(y_r, l_r)​$ without the need for an explicit combined loss.

---

> > ### Author Response · Authors · 2024-11-19
> > **Response to Reviewer BbUH (Part 2)**
> >
> > **Q2:** Impact of $\alpha$?
> >
> > **A2:** Thank you for your thoughtful feedback. In this context, alpha represents the decay factor of the running average, controlling how the average adopts to new data. We selected alpha of value 0.1, which is a commonly used value for running averages as it creates a stable trend while filtering out short-term variability. To provide more comparisons, we have included additional results for $\alpha = 1.0$ and $\alpha = 0.0$ in Appendix E.6, and Table 11 of the updated version of the paper. An $\alpha$ of 1.0 means that the running average equals the batch-wise average for each update. And $\alpha$ of 0.0 means that the running average is not updated and remains at its initialization value (in our case, it is the reward mean on the evaluation set when the reward model is trained). We have pasted the table here for your reference.
> >
> > ---
> > | $\alpha$ | MT-Bench | ECE $\downarrow$ (GSM8K) | AUC $\uparrow$ (GSM8K) | ACC $\uparrow$ (GSM8K) | ECE $\downarrow$ (SciQ) | AUC $\uparrow$ (SciQ) | ACC $\uparrow$ (SciQ) | ECE $\downarrow$ (CommonsenseQA) | AUC $\uparrow$ (CommonsenseQA) | ACC $\uparrow$ (CommonsenseQA) |
> > |---|---|---|---|---|---|---|---|---|---|---|
> > | 0 | 7.79 | 0.8833 | 0.5034 | 0.116 | 0.1056 | 0.5238 | 0.891 | 0.2178 | 0.5568 | 0.7649 |
> > | 0.1 | 8.05 | 0.8638 | 0.516 | 0.1031 | 0.0282 | 0.6513 | 0.904 | 0.1286 | 0.5621 | 0.7756 |
> > | 1.0 | 8.03 | 0.8827 | 0.5112 | 0.1145 | 0.0849 | 0.5493 | 0.907 | 0.1992 | 0.5632 | 0.7625 |
> >
> > ---
> >
> > | $\alpha$ | MT-Bench | ECE $\downarrow$ (TruthfulQA) | AUC $\uparrow$ (TruthfulQA) | ACC $\uparrow$ (TruthfulQA) | ECE $\downarrow$ (Object Counting) | AUC $\uparrow$ (Object Counting) | ACC $\uparrow$ (Object Counting) | ECE $\downarrow$ (Professional Knowledge) | AUC $\uparrow$ (Professional Knowledge) | ACC $\uparrow$ (Professional Knowledge) |
> > |---|---|---|---|---|---|---|---|---|---|---|
> > | 0 | 7.79 | 0.5185 | 0.5655 | 0.4394 | 0.4948 | 0.498 | 0.505 | 0.4753 | 0.5119 | 0.5024 |
> > | 0.1 | 8.05 | 0.4426 | 0.5303 | 0.4431 | 0.4839 | 0.5178 | 0.503 | 0.3949 | 0.4902 | 0.502 |
> > | 1.0 | 8.03 | 0.4965 | 0.5595 | 0.4333 | 0.4797 | 0.5011 | 0.52 | 0.4614 | 0.4968 | 0.4935 |
> > ---
> >
> > To summarize the results here:
> > * when $\alpha = 1.0$, we observe a significant decline in calibration performance, except for the BBH dataset. Similarly, when $\alpha = 0.0$, the performance is also worse than with $\alpha = 0.1$, both in terms of calibration performance and MT-Bench scores.

---

> > > ### Author Response · Authors · 2024-11-19
> > > **Response to Reviewer BbUH (Part 3)**
> > >
> > > **Q3:** Are you assuming $\hat{r}(y_c) > \Delta r$ and $\hat{r}(y_l) > \Delta r$?
> > >
> > > **A3:** Thank you for your comment. For current PPO-C, we indeed assume $\hat{r}(y_c) > \Delta r$ and $\hat{r}(y_l) > \Delta r$ and form a threshold-based approach.
> > > We also recognize that using a threshold-based approach may not be optimal. As a result, we explore an alternative difference-based approach that uses the difference between the current reward value and the running average of reward $(\hat{r_i} - \Delta r_t)$, and define the reward model adjustment as: $r_i = \hat{r_i} + (\hat{r_i} - \Delta r_t) * (s_i - 0.5)$. Our intuition is to use a smoother learning signal so that the reward adjustment scales with the relative deviation from the average. The results of this comparison are detailed in Appendix E.9 and Table 14 of the updated PDF, which we have included below for your reference:
> > >
> > > ---
> > > | Method      | MT/Arena-Hard | ECE ↓ (GSM8K) | AUC ↑ (GSM8K) | ACC ↑ (GSM8K) | ECE ↓ (SciQ) | AUC ↑ (SciQ) | ACC ↑ (SciQ) | ECE ↓ (CommonsenseQA) | AUC ↑ (CommonsenseQA) | ACC ↑ (CommonsenseQA) |
> > > |-------------|---------------|---------------|---------------|--------------|--------------|--------------|--------------|------------------------|-----------------------|-----------------------|
> > > | Threshold   | 8.05 / 14.1  | 0.8638        | 0.516         | 0.1031       | 0.0282       | 0.6513       | 0.904        | 0.1286                 | 0.5621                | 0.7756                |
> > > | Difference  | 7.91 / 13.8  | 0.8238        | 0.4937        | 0.119        | 0.0087       | 0.578        | 0.898        | 0.1153                 | 0.585                 | 0.7625                |
> > >
> > > ---
> > >
> > > | Method      | MT/Arena-Hard | ECE ↓ (TruthfulQA) | AUC ↑ (TruthfulQA) | ACC ↑ (TruthfulQA) | ECE ↓ (Object Counting) | AUC ↑ (Object Counting) | ACC ↑ (Object Counting) | ECE ↓ (Professional Knowledge) | AUC ↑ (Professional Knowledge) | ACC ↑ (Professional Knowledge) |
> > > |-------------|---------------|---------------------|--------------------|-------------------|-------------------------|-------------------------|-------------------------|-------------------------------|-------------------------------|-------------------------------|
> > > | Threshold   | 8.05 / 14.1  | 0.4426              | 0.5303             | 0.4431            | 0.4839                  | 0.5178                  | 0.503                  | 0.3949                        | 0.4902                        | 0.502                         |
> > > | Difference  | 7.91 / 13.8  | 0.4104              | 0.515              | 0.4492            | 0.4774                  | 0.5118                  | 0.496                  | 0.383                         | 0.509                         | 0.4902                        |
> > > ---
> > >
> > > To summarize:
> > > * The difference-based approach demonstrates improved calibration results, evidenced by a lower ECE, when compared to the threshold-based approach.
> > > * However, this improvement comes with a trade-off: a decrease in instruction-following capabilities, as reflected by lower scores in MT-Bench and Arena-Hard evaluations. Furthermore, accuracy on 4 out of 6 datasets shows a slight decline with the difference-based approach.

---

> > > > ### Author Response · Authors · 2024-11-19
> > > > **Response to Reviewer BbUH (Part 4)**
> > > >
> > > > **Q4:** The methods appear to rely heavily on heuristic rather than principled algorithm design.
> > > >
> > > > **A4:** Thank you for your concerns. While our methods appear straightforward, we have conducted extensive evaluations that demonstrate their robustness and generalizability across different dimensions:
> > > > * Generalization across model families, benchmarks, and RLHF training paradigms.
> > > >    * Our methods are successfully applied to two model families (Llama3-8B and Mistral-7B)
> > > >    * Our methods consistently improve model calibration and maintaining accuracy across six benchmark datasets in different domains, without compromising instruction-following capabilities (MT-Bench and Arena-Hard).
> > > >    * Our methods can be extended to both PPO and DPO (Sec 5.2 and Appendix E.11) training paradigms.
> > > > * Effectiveness of method design
> > > >    * Comparison against PPO$\dagger$ (using vanilla reward model in PPO training) in Table 1 demonstrates the effectiveness of using our calibrated reward model in PPO-M.
> > > >    * Additional analysis of different reward adjustment factors in PPO-C between our original threshold-based approach and the new difference-based approach (Table 14) reveals a trade-off between model calibration and instruction-following capabilities.
> > > > * Comprehensive hyperparameter studies
> > > >    * The impact of parameter $w$ and $\alpha$ for PPO-C (Table 10 and Table 11).
> > > >    * The impact of system prompt percentage (Table 12).
> > > >    * The impact of combining equations (1) and (2) for calibrated reward model training (Table 13).
> > > >
> > > > Collectively, these positive results across diverse evaluation settings offer evidence supporting the effectiveness and generalizability of our proposed methods. We believe that this justifies the contributions of our approach beyond heuristic, demonstrating both robustness and practical value.
> > > >
> > > > ---
> > > >
> > > > **Q5:** Why do PPO-M and PPO-C often result in lower ACC?
> > > >
> > > > **A5:** Thank you for your question. Here is a summary of the performance comparisons:
> > > > * **Llama-3-8B with Direct Answer (DA) prompting:** PPO-M outperforms PPO on 4 out of 6 datasets (GSM8K, CommonsenseQA, TruthfulQA, Object Counting), while PPO-C outperforms PPO on 2 out of 6 datasets (CommonsenseQA, Object Counting).
> > > > * **Llama-3-8B with Chain-of-Thought (CoT) prompting:** PPO-M outperforms PPO on 3 out of 6 datasets (GSM8K, TruthfulQA, Object Counting), while PPO-C outperforms PPO on 5 out of 6 datasets (GSM8K, SciQ, CommonsenseQA, TruthfulQA, Object Counting).
> > > >
> > > > For cases where PPO-M and PPO-C have lower accuracy scores than PPO, the discrepancy is usually less than 1 percentage point, while we can substantially improve model calibration: PPO-M and PPO-C largely reduce ECE by up to 5.23 (0.1729 to 0.1206) / 4.43 (0.1729 to 0.1286) percentage points with Direct-Answer prompting, and 7.88 (0.3651 to 0.2863) / 11.60 (0.3651 to 0.2491) percentage points with Chain-of-Thought prompting on six benchmarks for Llama-3.
> > > > The only notable accuracy drop is with Llama-3 on Professional Knowledge using CoT promoting, where even PPO with confidence-query prompt (PPO$\dagger$) performs worse in terms of accuracy.
> > > >
> > > > These results indicate that our proposed methods generally maintain competitive accuracy scores while improving model calibration substantially. We believe that this is a favorable trade-off, as well-calibrated models provide more reliable uncertainty estimates in real-world applications.

---

> > > > > ### Comment · Reviewer_BbUH · 2024-11-19
> > > > >
> > > > > Thanks for the comprehensive response and appreciate that you have added experiments for every question!
> > > > > I still have some remaining questions:
> > > > > 1. If the model has a similar capability to distinguish between chosen and rejected responses without explicitly training for $E[\log\sigma(R(x,y_c)-R(x,y_r))]$, what would be the reason? Is it maybe because of the inductive bias of preferring $h$ over $l$?
> > > > > 2. It seems that $\alpha$ is indeed a hyperparameter that needs to be tuned. Moreover, is the assumption that $\hat{y_c}>\Delta r$ and $\hat{y_l}<\Delta r$ general to all cases?  Could there be some $y_c$ with a reward lower than the mean reward of all training data? For example, in this case of $$[R(y_{c_1})=1,R(y_{r_1})=0.1,R(y_{c_2})=0.2,R(y_{r_2})=0.1],$$ the mean reward is $\Delta r=0.35$ and $R(y_{c_2})=0.2<0.35<\Delta r$.
> > > > >
> > > > > While I still regard these as heuristic methods lacking rigorous design and theoretical justification, I greatly appreciate the authors' efforts in demonstrating strong empirical results that may offset this limitation. To be honest, this could be a strong paper for conferences like EMNLP. However, given the standards of ICLR, I am inclined to maintain my current ratings, with the possibility of lowering my confidence level based on discussions with other reviewers.

---

> > > > > > ### Author Response · Authors · 2024-11-25
> > > > > > **Response to Reviewer BbUH (Part 1)**
> > > > > >
> > > > > > Thank you once again for your valuable comments and for recognizing the strength of our empirical results. Below, we provide detailed responses to your questions:
> > > > > >
> > > > > > **Q1:** If the model has a similar capability to distinguish between chosen and rejected responses without explicitly training, what would be the reason?
> > > > > >
> > > > > > **A1:**  Thank you for your question. We appreciate your careful examination and summarize two key reasons as follows.
> > > > > >    * **Initial checkpoint has the ability to separate chosen and rejected responses:** In our experiments, we fine-tune an existing reward model that is already trained to distinguish chosen and rejected pairs. The existing reward model, serving as the initial checkpoint of our calibrated reward model training, can already separate chosen and rejected responses by a large margin.
> > > > > >    * **Constructed calibration training set preserves the model separation capability:** In our calibration training dataset, the chosen and rejected responses can still be separable. This is because chosen responses are connected with high confidence, preferred over low confidence; and rejected responses are connected with low confidence, preferred over high confidence. Therefore, this **association between response quality and confidence level** helps the reward model preserve the capability to distinguish response quality and simultaneously learn appropriate confidence levels.
> > > > > >
> > > > > >
> > > > > > ---
> > > > > > **Q2:** It seems $\alpha$ needs to be tuned. Is the assumption that $\hat{y_c} > \Delta r$ and $\hat{y_r} < \Delta r$ general to all cases? Could there be some $y_c$ with a reward lower than the mean reward of all training data?
> > > > > >
> > > > > > **A2:** Thanks for your concern. Our choice of $\alpha = 0.1$ follows established practices in calculating exponential moving average. For example, in Adam optimizer [1], the first moment estimation (Algorithm 1 on page 2) is updated by $m_t = \beta_1 * m_{t-1} + ( 1 - \beta_1) * g_t$, where $\beta_1$ is set to $0.9$. The $\beta_1 = 0.9$ in Adam is equivalent to $(1 - \alpha)$ in our moving average calculation where $\alpha = 0.1$, which both serve as the decaying rate for the variable at timestep $t-1$. This indicates that $\alpha = 0.1$ is a validated choice in established practices. Our hyperparameter study ($\alpha=0.0$ and $\alpha=1.0$) in the previous reply shows that under extreme cases, the results still fall into a reasonable range of values.
> > > > > >
> > > > > > Regarding your question and counter-example on $\hat{y_c} > \Delta r$ and $\hat{y_r} < \Delta r$, we would like to clarify that PPO-C is used for policy model optimization, not reward model training. Chosen / rejected pairs are only used for training a reward model. In standard PPO, there are no chosen / rejected pairs, and PPO-C is dealing with individual responses sampled from the policy model. We use moving average as a dynamic reference point to measure the current policy model capability during PPO training. If the current response’s reward is much higher than the dynamic reference point, which indicates that the current response has a relatively higher quality than recent model responses, the response’s reward score will be increased if followed by a high confidence.
> > > > > >
> > > > > > [1] Kingma, Diederik P. and Jimmy Ba. “Adam: A Method for Stochastic Optimization.” ICLR 2015

---

> > > > > > > ### Author Response · Authors · 2024-11-25
> > > > > > > **Response to Reviewer BbUH (Part 2)**
> > > > > > >
> > > > > > > **Q3:** Heuristic methods lacking rigorous design and theoretical justification.
> > > > > > >
> > > > > > > **A3:** Thank you for your comment. We acknowledge that our methods are empirically-driven. While we truly appreciate the importance of theoretical rigor and understand your concern, we believe that heuristic approaches, after thorough validation on a wide range of settings, can also improve machine learning research. Several recent papers accepted to ICLR include carefully designed heuristic metrics in their methodology, for example:
> > > > > > >    * [1] proposes a Min-K probability approach (Eq. 1 in Sec. 3) for detecting pre-training data for black-box LLMs
> > > > > > >    * [2] proposes contrasting log probabilities between transformer layers in decoding (Sec 2.3) to improve factuality of LLMs
> > > > > > >    * [3] proposes an approach to aggregate specific attention heads (Eq. 3 - Eq. 5 in Sec 2.3) as “function vectors” to improve zero-shot task inference.
> > > > > > >
> > > > > > > Although these recent ICLR papers have heuristic designs and do not include rigorous theoretical justification, they all conduct thorough empirical evaluations and turn out to have high impacts in their respective research areas.
> > > > > > >
> > > > > > > Additionally, the RLHF framework itself indeed relies on large amounts of empirical training techniques, as suggested by a recent work [4] which identifies 25 useful empirical details that are necessary for effective RLHF implementation, indicating that empirical insights are also important for this research field.
> > > > > > >
> > > > > > > Therefore, we believe that our strong empirical results generalized across various models, benchmarks, RLHF training paradigms, and evaluation metrics demonstrate that our proposed approach can be a meaningful contribution to the field, especially with the growing importance of LLM trustworthiness and reliability.
> > > > > > >
> > > > > > > [1] Shi, Weijia, et al. “Detecting Pretraining Data From Large Language Models.” ICLR 2024.
> > > > > > >
> > > > > > > [2] Chuang, Yung-Sung, et al. “DoLa: Decoding by Contrasting Layers Improves Factuality in Large Language Models”. ICLR 2024.
> > > > > > >
> > > > > > > [3] Todd, Eric, et al. “Function Vectors in Large Language Models”. ICLR 2024.
> > > > > > >
> > > > > > > [4] “The N+ Implementation Details of RLHF with PPO: A Case Study on TL;DR Summarization”. COLM 2024.
> > > > > > >
> > > > > > > ---
> > > > > > > Thank you once again for your valuable feedback that helps us a lot on improving the manuscript! If you have additional concerns, please let us know and we are happy to address them.

---

> > > > > > > > ### Comment · Reviewer_BbUH · 2024-11-26
> > > > > > > >
> > > > > > > > Thanks very much for the detailed explanations. I now have a better understanding of the implementation details. However, I am inclined to maintain my current ratings due to only empirical-driven design, so I will keep my current rating. Because I am less sure whether this kind of empirical research is above the bar of ICLR, I will lower my confidence.
> > > > > > > >
> > > > > > > >
> > > > > > > > ***
> > > > > > > > Thanks again for being friendly and providing comprehensive answers from the beginning.

---

### Official Review · Reviewer_sUqe · 2024-10-31

**Soundness:** 2
**Presentation:** 3
**Contribution:** 3
**Rating:** 6
**Confidence:** 5

**Summary:**

The paper tackles the problem of calibration in LLMs that went through RLHF, and specifically calibration through verbalized confidence. Based on prior work that showed that RLHF hurt the model calibration, the authors point to the reward model as the source of the problem. They demonstrate that off-the-shelf reward model have a bias towards answers with high-confidence. As a solution, they propose two different methods that should overcome this problem - one focus on the reward model training and the other fix the rewards given during policy optimization.

**Strengths:**

- The paper has a clear structure, beginning with analysis experiments, moving to the proposed method, and concluding with experiments that showcase its effectiveness.
- The experiments are comprehensive, covering multiple datasets and showing clear improvement in calibration metrics.
- The problem addressed is relevant and of interest to the community.

**Weaknesses:**

- The conclusion drawn from Figure 2 doesn’t align with the plots. It’s not that the model after RLHF isn’t overconfident, but rather that the model before RLHF is also overconfident. Nearly all the bins fall below the diagonal line, indicating overconfidence. Additionally, why are the data points so sparse? How many prompts were used in this experiment?

- In Figure 3, since the reward models haven’t been trained with this type of prompt, I would suggest running a control experiment—appending the prompt with a random confidence score to both chosen and rejected answers (similar to PPO-M training). With random confidence scores, we would expect the reward model, on average, to still prefer the chosen answer. If it doesn’t, this would indicate that the prompt is so far out of distribution that the reward model isn’t applicable (which highlights its brittleness, though that’s a separate issue).

- While I understand the algorithm used in PPO-C, the intuition behind it isn’t clear to me. Why would the average reward across the dataset serve as a threshold for determining if a model’s response is correct? Couldn’t it be that most responses are either correct or incorrect, depending on the dataset and model? This approach seems like a heuristic that would only work on datasets where the model is correct about 50% of the time.
Additionally, why is $\Delta r$ defined as a running average of $r$ rather than $\hat{r}$? On a related note, wouldn’t an off-the-shelf reward model struggle to assign rewards to answers that include confidence scores (i.e., $r$) since it wasn’t trained on such data?

**Questions:**

- Regarding PPO-M, is it intended to be an additional objective to equation 1? Because there is nothing in equation 2 that will make the reward of the chosen answer higher than the rejected one.

- Line 270: what do you mean by “biased response”? Why does asking for confidence create bias?

---

> ### Author Response · Authors · 2024-11-19
> **Response to Reviewer sUqe (Part 1)**
>
> Thank you for your constructive feedback and comments. We have incorporated the suggested changes into our updated manuscript and will address your concerns below. Please let us know if our response addresses your concerns.
>
> **Q1:** The conclusion drawn from Figure 2 does not align with the plots. Why are data points so sparse. How many prompts were used in this experiment?
>
> **A1:** We agree with your observation that pre-RLHF models also show some degree of overconfidence. However, our point is to show that RLHF extensively increases the overconfidence of LLMs. In this experiment, we use all samples from CommonsenseQA validation set (1221 prompts in total), and bin the output confidence scores into 20 intervals (0.05 range per interval). For each bin, we aggregate all responses in that confidence interval to calculate the bin accuracy (with the last bin merging confidence 0.95 and 1.0). Empty bins mean no confidence scores fall into those intervals. RLHF-trained models have more empty bins in the lower confidence range, indicating that they have a stronger bias towards expressing high confidence scores (darker bins mean more samples falling into those bins). In contrast, pre-RLHF models show a broader range of confidence distributions across multiple intervals. This suggests that RLHF tends to make LLMs more overconfident. We have made Figure 2 caption clearer in our revised version of the paper.
>
> ---
>
> **Q2:** Appending the prompt with a random confidence score to both chosen and rejected answers.
>
> **A2:** Thank you for your suggestion. We also recognize the potential concern that the reward model may be biased by the confidence-query system prompt, which is often included to enable the model to verbalize its confidence. We have conducted an analysis to evaluate the impact of this prompt on additional 15 reward models, as detailed in Appendix E.2, and Figures 17, 18, 19. The left panel of these figures presents the exact setting as our preliminary experiments, while the right panel explores a new setting on the same set of reward models where the **confidence-query prompt is removed**, and only random confidence scores are appended to the model responses.
> To summarize the findings:
> * When the confidence-query prompt is removed, under the confidence_reversed setting, where random high scores are appended to rejected responses and low scores to chosen responses, many models **continue to assign higher scores to the chosen responses**. This contrasts with cases where the confidence-query prompt is included.
> * Moreover, when comparing the same responses with high versus low confidence scores, we consistently observe a tendency for the reward models to prefer high-confidence responses regardless of their correctness, indicating that the bias persists.

---

> > ### Author Response · Authors · 2024-11-19
> > **Response to Reviewer sUqe (Part 2)**
> >
> > **Q3:** The intuition behind PPO-C isn’t clear. Why would the average reward across the dataset serve as a threshold for determining if a model’s response is correct? Why is $\Delta r$ defined as a running average of $r$ rather than $\hat{r}$?
> >
> > **A3:** Thank you for your thoughtful concern. We first apologize for any confusion caused and clarify that we indeed use $\hat{r}$ for the running average computation. The running average is calculated as the weighted sum of previous running average and the current batch mean of unbiased rewards, $\hat{r}$. This ensures that the reward model is applied to model responses it has been trained for, providing an unbiased estimation of response quality that is not affected by any confidence scores. We have updated the formula in the revised version of the paper.
> >
> > In addition, your concern about the relationship between average reward and correctness is well-taken, and our intention is not to assume a specific correctness rate, but to use the average reward as a dynamic reference to represent the current model capability. We agree that this threshold-based approach may not be optimal. As a result, we explore an alternative difference-based approach that uses the difference between the current reward value and the running average of reward $(\hat{r_i} - \Delta r_t)$, and define the reward model adjustment as: $r_i = \hat{r_i} + (\hat{r_i} - \Delta r_t) * (s_i - 0.5)$. Our intuition is to use a smoother learning signal so that the reward adjustment scales with the relative deviation from the average. The results of the comparison between threshold-based and difference-based approaches are detailed in Appendix E.9 and Table 14 of the updated PDF, which we have included below for your reference:
> >
> > ---
> > | Method      | MT/Arena-Hard | ECE ↓ (GSM8K) | AUC ↑ (GSM8K) | ACC ↑ (GSM8K) | ECE ↓ (SciQ) | AUC ↑ (SciQ) | ACC ↑ (SciQ) | ECE ↓ (CommonsenseQA) | AUC ↑ (CommonsenseQA) | ACC ↑ (CommonsenseQA) |
> > |-------------|---------------|---------------|---------------|--------------|--------------|--------------|--------------|------------------------|-----------------------|-----------------------|
> > | Threshold   | 8.05 / 14.1  | 0.8638        | 0.516         | 0.1031       | 0.0282       | 0.6513       | 0.904        | 0.1286                 | 0.5621                | 0.7756                |
> > | Difference  | 7.91 / 13.8  | 0.8238        | 0.4937        | 0.119        | 0.0087       | 0.578        | 0.898        | 0.1153                 | 0.585                 | 0.7625                |
> >
> > ---
> >
> > | Method      | MT/Arena-Hard | ECE ↓ (TruthfulQA) | AUC ↑ (TruthfulQA) | ACC ↑ (TruthfulQA) | ECE ↓ (Object Counting) | AUC ↑ (Object Counting) | ACC ↑ (Object Counting) | ECE ↓ (Professional Knowledge) | AUC ↑ (Professional Knowledge) | ACC ↑ (Professional Knowledge) |
> > |-------------|---------------|---------------------|--------------------|-------------------|-------------------------|-------------------------|-------------------------|-------------------------------|-------------------------------|-------------------------------|
> > | Threshold   | 8.05 / 14.1  | 0.4426              | 0.5303             | 0.4431            | 0.4839                  | 0.5178                  | 0.503                  | 0.3949                        | 0.4902                        | 0.502                         |
> > | Difference  | 7.91 / 13.8  | 0.4104              | 0.515              | 0.4492            | 0.4774                  | 0.5118                  | 0.496                  | 0.383                         | 0.509                         | 0.4902                        |
> > ---
> >
> > To summarize:
> > * The difference-based approach demonstrates improved calibration results, evidenced by a lower ECE, when compared to the threshold-based approach.
> > * However, this improvement comes with a trade-off: a decrease in instruction-following capabilities, as reflected by lower scores in MT-Bench and Arena-Hard evaluations. Furthermore, accuracy on 4 out of 6 datasets shows a slight decline with the difference-based approach.

---

> ### Author Response · Authors · 2024-11-19
> **Response to Reviewer sUqe (Part 3)**
>
> **Q4:** Regarding PPO-M, is it intended to be an additional objective to equation 1
>
> **A4:** Thank you for your question. Equation (2) is intended to function as a stand-alone objective. In our experiments, when **fine-tuning an existing reward model** on our calibration dataset, we did not observe any diminishment in the model’s ability to distinguish between chosen and rejected responses. As shown in Figure 6 of Section 4.2 in the updated PDF, we provide a direct comparison between our calibrated reward model and its pre-calibrated version. Specifically, in the answer_only row, which uses the original RewardBench datasets without modifications, **the chosen/rejected preference ratio remains largely unchanged**.
> Additionally, in Appendix E.8 and Figure 24, we present the training plots of the reward model trained using equations (1) and (2) in combination, compared to training with equation (2) alone. The model trained solely with equation (2) maintains a similar trend of distinguishing between chosen and rejected responses to the model trained with the combined loss.
>
> Furthermore, we conducted experiments comparing PPO-M using the reward model trained with equation (2) alone versus in combination with equation (1), as shown in Table 13 of the updated manuscript. For your convenience, we have provided the table below.
>
> ---
> | Loss                             | MT-Bench | ECE ↓ (GSM8K) | AUC ↑ (GSM8K) | ACC ↑ (GSM8K) | ECE ↓ (SciQ) | AUC ↑ (SciQ) | ACC ↑ (SciQ) | ECE ↓ (CommonsenseQA) | AUC ↑ (CommonsenseQA) | ACC ↑ (CommonsenseQA) |
> |----------------------------------|----------|---------------|---------------|--------------|--------------|--------------|--------------|------------------------|-----------------------|-----------------------|
> | (2)          | 8.05     | 0.8638        | 0.516         | 0.1031       | 0.0282       | 0.6513       | 0.904        | 0.1286                 | 0.5621                | 0.7756                |
> | (1) + (2) | 7.75     | 0.8891        | 0.4974        | 0.1107       | 0.1043       | 0.5186       | 0.894        | 0.2286                 | 0.528                 | 0.7584                |
>
> ---
>
> | Loss                             | MT-Bench | ECE ↓ (TruthfulQA) | AUC ↑ (TruthfulQA) | ACC ↑ (TruthfulQA) | ECE ↓ (Object Counting) | AUC ↑ (Object Counting) | ACC ↑ (Object Counting) | ECE ↓ (Professional Knowledge) | AUC ↑ (Professional Knowledge) | ACC ↑ (Professional Knowledge) |
> |----------------------------------|----------|---------------------|--------------------|-------------------|-------------------------|-------------------------|-------------------------|-------------------------------|-------------------------------|-------------------------------|
> | (1)           | 8.05     | 0.4426              | 0.5303             | 0.4431            | 0.4839                  | 0.5178                  | 0.503                  | 0.3949                        | 0.4902                        | 0.502                         |
> | (1) + (2) | 7.75     | 0.5006              | 0.564              | 0.4565            | 0.518                   | 0.5                     | 0.482                  | 0.4786                        | 0.4964                        | 0.5061                        |
> ---
>
> The results indicate that PPO-M using the reward model trained with the combined loss does not achieve better calibration results.
> Moreover, combining equations (1) and (2) would require additional computational resources since we need to process two extra samples. Given these observations, we have chosen to set equation (2) as a standalone loss.
>
> ---
>
> **Q5:** Line 270: What do you mean by “biased response”
>
> **A5:** Thank you for pointing this out. We mean that including a confidence score within the response can lead to a biased reward score (as shown in our preliminary study in Sec 2.2). To address this, we first parse and remove the confidence score to obtain a clean (unbiased) response for evaluation.

---

> > ### Comment · Reviewer_sUqe · 2024-11-19
> >
> > I thank the authors for their response and explanation. The fixes they did were necessary as I now have a different understanding of the proposed algorithms. Overall, I think the paper points to an important problem with RLHF and proposes a solution that empirically helps to reduce the calibration issues. However, the proposed algorithms are mainly heuristic, and I was not convinced that these are the right solutions to this problem. Therefore, I will raise my score but only to borderline accept.

---

> > > ### Author Response · Authors · 2024-11-25
> > > **Thank you for your comment!**
> > >
> > > Thank you very much for your constructive feedback! We are glad that most of the issues raised are addressed and you find the updates necessary. Thank you once again for your valuable suggestions to make our paper better!

---

### Official Review · Reviewer_i6YC · 2024-11-02

**Soundness:** 2
**Presentation:** 3
**Contribution:** 2
**Rating:** 8
**Confidence:** 3

**Summary:**

This paper focuses on LLM calibration: the alignment between the confidence a model expresses and its actual performance. Previous research has demonstrated the phenomenon of overconfidence in LLMs, especially those trained using RLHF, which tend to have a sharpened output probability. The systematic bias in reward models used for RLHF is one of the key causes of LLM overconfidence.

The paper proposes two methods to address the issue of verbalized overconfidence in LLMs trained with RLHF:

(1) **PPO-M**: a reward model re-training objective that aims to fine-tune existing BT-based reward models. PPO-M achieves this by augmenting existing pairwise ranking datasets with verbal confidence scores. The author filtered samples from various open-source datasets, prepended a confidence-query system prompt to each, and randomly assigned high and low confidence scores, creating a modified training dataset.

(2) **PPO-C**: a reward-shaping method that dynamically adjusts reward scores during PPO training based on a moving average of past rewards and the model's verbalized confidence level. PPO-C differs from vanilla PPO in that it incorporates a confidence-query system prompt into a portion of the training prompts to elicit verbal confidence scores from the model. The reward function promotes higher confidence for responses exceeding the moving average and lower confidence for those falling below.

---
*Nov. 26 update*: most of my concerns are resolved, satisfied with the added evaluations, and raised the rating from 6 ---> 8 while maintaining my confidence of 3.

**Strengths:**

- This paper is very well-written and easy to follow. The motivation is well-supported by evidence, including visualizations of confidence distributions and accuracy, that highlight the overconfidence phenomenon.
- The proposed methods are straightforward, yet backed by a sufficient amount of evaluation on different families of open-weights models and on diverse benchmarks. The thorough evaluation, along with the authors' transparency in presenting both the strengths and limitations of their approach, makes their findings trustworthy - small improvements in the factuality-focused benchmarks as expected (e.g. TruthfulQA benchmark) and almost no improvement in chat performance (e.g. MT-Bench, Arena-Hard).

**Weaknesses:**

- Referring to Eq. 2 as the "PPO-M" loss could be confusing, as it is not directly related to the standard PPO objective. This equation is, in fact, a re-training objective for the reward model rather than a modification of PPO itself. The labels "PPO-M" and "PPO-C" might suggest that the paper centers on enhancing PPO objectives, when the proposed methods primarily aim at co-training the reward model with verbal confidence. This distinction could benefit from clarification to prevent misunderstanding.

- While not introduced in this work, the use of verbal confidence to measure a model's confidence in its response may not be ideal. Language models, especially those under 70B parameters, may simulate confidence without true understanding due to limitations in reasoning capability. Have you examined the logits for the 0–10 range when asking for verbal confidence? Observing how these tokens are distributed might provide insight into whether the model is indeed producing a typical probability distribution or simply outputting values. Given these concerns, more concrete metrics, such as those derived from entropy in token-space, might be more accurate indicators of true confidence.

- The confidence metric presented in Figure 2 lacks clarity, and the "perfect calibration" line does not contribute meaningful differentiation between a model's overconfidence and cases where the dataset may simply be too easy. Maybe Llama-3-8B-PPO's performance on CommonsenseQA could indicate that the questions are too easy leading to high confidence in correct answers.

- Given that Figure 2 does not reflect the effects of PPO-M and PPO-C, it would be valuable to include a distribution of confidence scores after tuning with these proposed methods. This would enable a direct comparison of the effects of the calibration techniques on model confidence.

-  Figure 3 would benefit from results that combine the answer and confidence score to more clearly illustrate how the proposed methods affect the chosen/rejected ratio.

**Questions:**

- I understand that just for the sake of an evaluation of how the current chat models are conditioned on the given verbal confidence, appending a random confidence score to the RewardBench dataset may make sense. Why not simply prompt the chat models to self-annotate? Asking models to generate their own confidence scores could yield a more genuine assessment of how they internally gauge response certainty.

- Figure 2 primarily highlights instances of overconfidence, but are there scenarios where the SFT model consistently exhibits low confidence? It would be interesting to see if the proposed methods can help address underconfidence in addition to overconfidence, particularly if SFT models tend to understate confidence levels.

---

> ### Author Response · Authors · 2024-11-19
> **Response to Reviewer i6YC (Part 1)**
>
> Thank you for your valuable feedback. We have incorporated the feedback into our updated manuscript and will address your concerns below:
>
> **Q1:** The “PPO-M” loss could be confusing.
>
> **A1:** Thank you for raising the concern. To avoid any confusion, we have revised the terminology in the updated version. We rename the loss in Eq. (2) to “CRM” to better reflect its purpose in Calibrated Reward Modeling.
>
> ---
>
> **Q2:** The use of verbal confidence to measure a model’s confidence in its response may not be ideal.
>
> **A2:** Thank you for your comment. We recognize that using verbal confidence to measure model confidence, particularly for smaller models under 70B, may have limitations due to their constrained reasoning capabilities. Nonetheless, the use of 7B models is widely adopted in related works within this field [1][2][3]. In our study, we use verbalized confidence because it provides direct feedback in natural language to LLM users without access to logits, and previous studies [4][5] demonstrate that using verbalized confidence generally leads to lower expected calibration error than logit-based methods.
>
> In response to your query about examining the logits for the 0 to 10 range when evaluating verbal confidence, we have conducted further analysis. In Appendix E.5, Figure 23 of the updated PDF, we present density plots of log probability of number 0 to 10, specifically at the locations corresponding to confidence scores on responses for TruthfulQA dataset.
> To summarize the observations:
> * We observe that certain numbers demonstrate notably high density. For example, the number 10 exhibits high density for PPO, while the number 9 shows a similar behavior for PPO-M. The log prob for numbers are not uniformly distributed, which suggests that the model is not randomly generating number values but rather exhibits a tendency to favor certain numbers in its confidence outputs.
>
> [1] Xu, Tianyang, et al. "SaySelf: Teaching LLMs to Express Confidence with Self-Reflective Rationales." The 2024 Conference on Empirical Methods in Natural Language Processing.
>
> [2] Tao, Shuchang, et al. "When to Trust LLMs: Aligning Confidence with Response Quality." arXiv preprint arXiv:2404.17287 (2024).
>
> [3] Han, Haixia, et al. "Enhancing Confidence Expression in Large Language Models Through Learning from Past Experience." arXiv preprint arXiv:2404.10315 (2024).
>
> [4] Tian, Katherine, et al. "Just Ask for Calibration: Strategies for Eliciting Calibrated Confidence Scores from Language Models Fine-Tuned with Human Feedback." The 2023 Conference on Empirical Methods in Natural Language Processing.
>
> [5] Lin, Stephanie, Jacob Hilton, and Owain Evans. "Teaching Models to Express Their Uncertainty in Words." Transactions on Machine Learning Research.
>
> ---
>
> **Q3:** The confidence metric presented in Figure 2 lacks clarity.
>
> **A3:** Thank you for your concern. It is true that Llama3-8b achieves over 70% accuracy on CommonsenseQA. To provide a broader perspective, we have included results on additional datasets in Appendix E.1, and Figures 12 through 16, covering 5 other datasets. For example, on TruthfulQA (Figure 16), where the model achieves approximately 40% accuracy, we observe that RLHF-trained LLMs’ confidence tends to concentrate within high-confidence bins. In contrast, the pre-RLHF version exhibits a broader distribution of confidence scores. This suggests that observed confidence is not solely attributable to the simplicity of the dataset.
>
> ---
>
> **Q4:** It would be valuable to include a distribution of confidence scores after tuning with these proposed methods.
>
> **A4:** Thank you for your suggestion. We have added a comparison of the confidence distributions for PPO, PPO-M, DPO, and CDPO (an extension of PPO-M to DPO-based pipelines) in Appendix E.4 and Figure 22 of the updated version of our paper. To summarize our observations, we note that the confidence distribution for PPO-M is slightly shifted to the left compared to PPO, showing a decrease in high-confidence scores (e.g., confidence level 10, which represents a highly overconfident state) and an increase in lower confidence categories. For CDPO, this phenomenon is more obvious; compared to DPO, the confidence distribution of CDPO is more widely spread across categories, with a noticeable increase in lower-confidence levels.

---

> > ### Author Response · Authors · 2024-11-19
> > **Response to Reviewer i6YC (Part 2)**
> >
> > **Q5:** Figure 3 would benefit from results that combine the answer and confidence score to more clearly illustrate how the proposed methods affect the chosen/rejected ratio.
> >
> > **A5:** Thank you for the suggestion. In Figure 3, chosen_with_conf, rejected_with_conf, and confidence_reversed integrate the answers with confidence scores is to demonstrate the systematic biases present in existing reward models (Figures for additional reward models can be found in Figures 16-18 in Appendix Section E.2). To further address your inquiry and demonstrate the effectiveness of our proposed methods, we have updated Figure 6 in Section 4.2 to include a direct comparison between the pre-calibrated Llama3-8b-rm and our calibrated version, Llama3-8b-crm, accompanied by new analyses presented in the updated version.
> > To summarize:
> > * The chosen and rejected ratio on the original model responses without appended confidence scores does not exhibit a significant difference. However, when evaluated on the rejected responses with high and low confidence scores, the pre-calibrated version consistently favors high-confidence responses. In contrast, our calibrated reward model tends to favor responses with low confidence - a behavior we aim to achieve.
> >
> > ---
> >
> > **Q6:** Why not simply prompt the chat models to self-annotate?
> >
> > **A6:** Thank you for your comment. The RewardBench experiments are specifically designed to demonstrate the systematic biases present in existing reward models across the entire confidence range, providing supporting evidence for the potential root cause of overconfidence in RLHF-trained LLMs. The primary focus of this experiment is to evaluate the behavior and biases of **reward models**, rather than of **chat models**.
> >
> > In our main results, we indeed prompt the model to verbalize their own confidence in their generated responses, which offers a more genuine assessment of how they internally gauge response certainty.
> >
> > ---
> >
> > **Q7:** Are there scenarios where the SFT models consistently exhibit low confidence?
> >
> > **A7:** Thank you for your valuable insight. This could be a possible scenario for SFT models in specific domains, while in our experiments, we did not observe scenarios where SFT models consistently exhibit low confidence. However, our method is fundamentally designed to align confidence scores with response quality, meaning it can generalize to scenarios of low confidence as well. By using a calibrated reward model and applying PPO, SFT models that initially exhibit low confidence should achieve improved calibration following RLHF.

---

> ### Comment · Reviewer_i6YC · 2024-11-26
> **Thank you**
>
> Thank you for the effort you put into the rebuttal. I am generally satisfied with the additional evaluations and the improvements in the paper's clarity, therefore I will improve my rating on this paper toward acceptance.
>
> ---
> *Quick question*:
>
> I appreciate the making of Figure 23 in Appendix E.5. The figures look good, though they were slightly different from what I had in mind at the time. If I understood correctly, each logit distribution corresponds to a subset of QA that has been given a particular confidence score by the model, which means the size of the subset could vary significantly across different confidence scores and I can foresee that lower confidence score will be a much lower amount of samples. If we can get a sense of how many QAs are used to construct every distribution that would be helpful statistically speaking.
>
> I was thinking about the logit distribution over the verbal confidence tokens for a single QA. Concretely, $\rm{P}(\rm{confidence}|x_{t<i}), \rm{confidence} \in [<0>,<1>, ...,<10>]$. I want to see if the model truly understands rating confidence as a smooth single-modal curve centered around the confidence score with the highest probability, and I also would like to learn if you believe that this figure would be helpful or not.

---

> ### Author Response · Authors · 2024-12-01
> **Thank you for your comment!**
>
> We sincerely thank you for your comments and recognition of our work and additional evaluations.
>
> Regarding your question, since it is not permissible to upload new figures at this moment, we will provide the statistics here and include the plot in the camera-ready version.
>
> For the current Figure 23, it incorporates all 817 samples from the TruthfulQA dataset. We extracted the log probabilities for 10 numbers from each sample and aggregated them to produce the figure.
>
> To demonstrate the distribution for a single QA sample, we randomly select one sample from the TruthfulQA dataset (round to 4 digits for clarity here):
>
> For PPO:
>    * {1: 1.7925e-06, 2: 1.9081e-06, 3: 5.1868e-06, 4: 1.9271e-05, 5: 0.0004, 6: 0.0020, 7: 0.0348, 8: 0.2575, 9: 0.3746, 10: 0.3306}
>
> For PPO-M:
>    * {1: 1.0372e-07, 2: 7.5881e-08, 3: 1.9377e-07, 4: 9.8404e-07, 5: 2.1040e-05, 6: 0.0002, 7: 0.0096, 8: 0.1705, 9: 0.7641, 10: 0.0553}
>
> To summarize, we observe that both distributions are unimodal, with PPO-M being more concentrated than standard PPO. This aligns with our expectations, as standard PPO is not explicitly optimized for outputting verbalized confidence scores. Overall, we agree that this visualization is a helpful tool for understanding model behavior in assigning confidence scores, and we appreciate your suggestion to include it.
>
> ---
>
> Thank you once again for your valuable feedback that makes our paper better.

---

### Official Review · Reviewer_Wbvf · 2024-11-03

**Soundness:** 4
**Presentation:** 4
**Contribution:** 2
**Rating:** 6
**Confidence:** 4

**Summary:**

In instruction-following for language models, the paper studies the task of reporting a confidence score in the helpfulness and accuracy of the model's own response. The paper shows that existing reward models are biased in favor of high-confidence scores, and proposes two mitigation methods. The first method involves data augmentation for the reward model, and the second method adds a customized term to the reward in PPO. The methods show improved calibration with no noticeable performance degradation.

**Strengths:**

The paper studies an important and interesting topic, that of verbalized calibration in instruction-following for language models. It is easy to follow and clearly demonstrates the existence of the problem and the success of the proposed mitigation methods. The methods do not require new data to be collected, making them straightforward to use, and yet the improved calibration transfers from reward model scores to accuracy on multiple-choice datasets.

**Weaknesses:**

Although the paper studies a broadly important topic and is well-executed, I think the strength of the contribution itself is somewhat limited:
- From a scientific perspective, the methods do not offer much insight: there is no distinction between different confidence levels other than "high scores" and "low scores", and the methods just increase the reward for high scores for high quality responses and low scores for low quality responses. So it is not really surprising that they improve calibration.
- This wouldn't be a problem if the methods had a lot of practical value. But from a practical perspective, the methods don't seem that useful either: if the user wanted a confidence score in the quality of the response, you could just display the reward model score (or some prompt-calibrated version) to the user. It's true that it's nice to have this distilled into the model, but there are many alternative ways in which one could do this, such as distilling the reward model score into a special token's logit. And if you just want a calibrated confidence score in the accuracy of a short-form response, there are many existing alternatives available.

In spite of this weakness, I consider the paper to be above the acceptance threshold because of the careful analysis the paper provides.

I also have a more minor issue with the definition of gamma for the PPO-C method (Line 279): the factor of |r_i (hat)| doesn't make sense to me. Unless the reward model is trained with an L2 penalty on the rewards or similar, the location of the reward is meaningless, since the loss uses only the difference between rewards, so taking an absolute value doesn't make sense. Even if the reward model is trained with an L2 penalty, I don't see why there should be a bigger adjustment for rewards that are far from 0 – or if there is some reason, it may be good to discuss it. Instead, to ensure that gamma is correctly scaled, it would make more sense to me to use some measure of spread of the rewards, such as standard deviation.

A couple typos:
- Line 134: "Dand" -> "D and"
- Line 1046: "calirbation" -> "calibration"

**Questions:**

I didn't understand the paragraph at Lines 239-244. It sounds like you are saying you didn't train the reward model yourself for PPO-M, but I don't see how that could be, since you need a reward model trained on your new calibration dataset. Are you saying that you fine-tuned an existing reward model on the new calibration dataset? Please clarify.

---

> ### Author Response · Authors · 2024-11-19
> **Response to Reviewer Wbvf (Part 1)**
>
> Thank you very much for your valuable comments. We have revised the manuscript to incorporate your suggestions, and we will address your concerns in detail below:
>
> **Q1:** From a scientific perspective, the method does not offer much insight.
>
> **A1:** We thank the reviewer for the concern regarding the insights provided by our work. To clarify, we summarize the key insights offered below:
> * **Identification of overconfidence root cause:** we identify the potential root cause of overconfidence in RLHF-trained LLMs and demonstrate how reward model biases can propagate through RLHF, leading to overconfident behavior in LLMs, which hasn’t been recognized by previous studies.
> * **Effective bias mitigation through simple intervention:** We show that simple intervention of reward scores on binary preferences can already effectively address the reward model bias.
>
> We agree that more fine-grained confidence levels could potentially lead to better calibration. However, this would require k-wise preference datasets, while many current preference learning datasets are based on binary human preferences [1][2][3].
> We have already shown that binary distinctions can already improve model calibration, and we leave more fine-grained levels of confidence to future studies.
>
> [1] Wang, Zhilin, et al. "HelpSteer2-Preference: Complementing Ratings with Preferences." arXiv preprint arXiv:2410.01257 (2024).
>
> [2] Bai, Yuntao, et al. "Training a helpful and harmless assistant with reinforcement learning from human feedback." arXiv preprint arXiv:2204.05862 (2022).
>
> [3] Ji, Jiaming, et al. "PKU-SafeRLHF: Towards Multi-Level Safety Alignment for LLMs with Human Preference." arXiv preprint arXiv:2406.15513 (2024).
>
> ---
>
> **Q2:** From a particle perspective, the methods don’t seem that useful either
>
> **A2:** Thank you for raising the concerns. While we appreciate your perspective, we respectfully disagree with the argument. Reward scores can be challenging to interpret due to different scales across prompts, as illustrated in Figure 1, and are typically compared in a pairwise manner. These scores are generally designed to measure the quality of model responses rather than assess model confidence. Furthermore, reward models are not consistently available in real-world applications. For example, when users interact with LLM chatbots, it is often more practical to integrate confidence calibration directly into the model generation process rather than relying on external scoring models. Additionally, verbalized confidence in LLMs has become an active area of research, with prior studies demonstrating the effectiveness of verbalized confidence over logit-based scores. While overconfidence phenomena have been observed, they remain unexplained [1] [2].
>
> From a practical standpoint, our proposed methods offer several key advantages:
> * **Seamless integration:** They can be integrated into existing PPO or DPO pipelines, providing better calibration results without compromising model performance on downstream tasks or reducing instruction-following capabilities compared to standard PPO or DPO.
> * **No “golden” labels required:** Unlike previous training-based methods for improving model calibration [3][4] that require “golden” confidence labels for supervised fine-tuning, our proposed calibration training simply augments existing pairwise datasets with randomly assigned confidence scores.
> * **Efficiency:** While PPO-M involves the calibration of a reward model, PPO-C introduces no additional computational overhead compared to standard PPO.
>
> Therefore, we believe our proposed methods hold practical values due to its ability to enhance calibration seamlessly, maintain performance and efficiency, and eliminate the need for label requirements.
>
> [1] Tian, Katherine, et al. "Just Ask for Calibration: Strategies for Eliciting Calibrated Confidence Scores from Language Models Fine-Tuned with Human Feedback." The 2023 Conference on Empirical Methods in Natural Language Processing.
>
> [2] Xiong, Miao, et al. "Can LLMs Express Their Uncertainty? An Empirical Evaluation of Confidence Elicitation in LLMs." The Twelfth International Conference on Learning Representations.
>
> [3] Lin, Stephanie, Jacob Hilton, and Owain Evans. "Teaching Models to Express Their Uncertainty in Words." Transactions on Machine Learning Research.
>
> [4] Han, Haixia, et al. "Enhancing Confidence Expression in Large Language Models Through Learning from Past Experience." arXiv preprint arXiv:2404.10315 (2024).

---

> > ### Author Response · Authors · 2024-11-19
> > **Response to Reviewer Wbvf (Part 2)**
> >
> > **Q3:** Issue with the definition of gamma for PPO-C
> >
> > **A3:** Thank you for your comment. In our study, we utilize the standard reward model where the reward is represented as a scalar value. The value is normalized during computation, a common technique in current PPO pipelines. We acknowledge that using the absolute value of the reward as a scaling factor gamma may not be ideal. Considering that reward models are trained to capture relative preference, we address this by conducting an additional experiment that uses the difference between the current reward value and the running average of reward $(\hat{r_i} - \Delta r_t)$ as the scaling factor for adjustment. Specifically, the adjusted reward score is expressed as $r_i = \hat{r_i} + (\hat{r_i} - \Delta r_t) * (s_i - 0.5)$. The results of this experiment are detailed in Appendix E.9 and Table 14 of the updated PDF, which we have also pasted below for your reference:
> >
> > ---
> > | Method      | MT/Arena-Hard | ECE ↓ (GSM8K) | AUC ↑ (GSM8K) | ACC ↑ (GSM8K) | ECE ↓ (SciQ) | AUC ↑ (SciQ) | ACC ↑ (SciQ) | ECE ↓ (CommonsenseQA) | AUC ↑ (CommonsenseQA) | ACC ↑ (CommonsenseQA) |
> > |-------------|---------------|---------------|---------------|--------------|--------------|--------------|--------------|------------------------|-----------------------|-----------------------|
> > | Threshold   | 8.05 / 14.1  | 0.8638        | 0.516         | 0.1031       | 0.0282       | 0.6513       | 0.904        | 0.1286                 | 0.5621                | 0.7756                |
> > | Difference  | 7.91 / 13.8  | 0.8238        | 0.4937        | 0.119        | 0.0087       | 0.578        | 0.898        | 0.1153                 | 0.585                 | 0.7625                |
> >
> > ---
> >
> > | Method      | MT/Arena-Hard | ECE ↓ (TruthfulQA) | AUC ↑ (TruthfulQA) | ACC ↑ (TruthfulQA) | ECE ↓ (Object Counting) | AUC ↑ (Object Counting) | ACC ↑ (Object Counting) | ECE ↓ (Professional Knowledge) | AUC ↑ (Professional Knowledge) | ACC ↑ (Professional Knowledge) |
> > |-------------|---------------|---------------------|--------------------|-------------------|-------------------------|-------------------------|-------------------------|-------------------------------|-------------------------------|-------------------------------|
> > | Threshold   | 8.05 / 14.1  | 0.4426              | 0.5303             | 0.4431            | 0.4839                  | 0.5178                  | 0.503                  | 0.3949                        | 0.4902                        | 0.502                         |
> > | Difference  | 7.91 / 13.8  | 0.4104              | 0.515              | 0.4492            | 0.4774                  | 0.5118                  | 0.496                  | 0.383                         | 0.509                         | 0.4902                        |
> > ---
> > To summarize:
> > * Compared to the threshold-based approach, the difference-based approach demonstrates better calibration results, as evidenced by a lower ECE.
> > * However, this improvement comes at the cost of a decrease in instruction-following capabilities as shown by lower scores in MT-Bench and Arena-Hard evaluations. Additionally, 4 out of 6 datasets show a slight decrease in accuracy with the difference-based approach. We hypothesize that this is because in most of our reward training, the reward score is close to 0-centered, so the absolute value of reward still makes sense, and the difference of $\hat{r_i}$ and $\Delta r_t$ might lead to smaller reward signals.
> >
> > ---
> >
> > **Q4:** A couple typos, Line 134: "Dand" -> "D and" and Line 1046: "calirbation" -> "calibration".
> >
> > **A4:** Thank you for pointing out the typos, we have corrected them in the updated version.
> >
> > ---
> >
> > **Q5:** I didn’t understand lines at 239-244.
> >
> > **A5:** Thank you for requesting clarification. Yes, you are correct; we fine-tune an existing reward model on the calibration dataset using our proposed loss function, rather than training a new reward model from scratch. To avoid confusion, we have added a clarification sentence in the updated version at line 240 to 241.

---

> ### Comment · Reviewer_Wbvf · 2024-11-20
> **Response to rebuttal**
>
> Thank you for your rebuttal. The new version of PPO-C makes more sense to me. The new experiment is interesting, but did you forget to include the scaling factor of $w = 0.5$, i.e. would it not be more comparable to take the reward to be $r_i=\hat r_i+w*(\hat r_i-\Delta r_t)*(s_i-0.5)$ with $w=0.5$? Failing to include this scaling factor would make the reward adjustment twice as large, which may explain why the new version favors calibration over instruction-following. Perhaps the running average of reward is close to 0, making the methods almost identical anyway (modulo the scaling factor issue). If so, I'd suggest using the new version, as it makes more theoretical sense, even though the two may be almost identical in practice.

---

> > ### Author Response · Authors · 2024-11-25
> > **Response to Reviewer Wbvf**
> >
> > Thank you for your valuable feedback! Based on your suggestions, we have updated our manuscript accordingly and uploaded the revised version. All changes are highlighted in orange. To summarize:
> >    * We have updated PPO-C to the difference-based approach in our main text, as shown in Figure 5 and last lines of Tables 1 and 2.
> >    * We have conducted hyperparameter sensitivity experiments for the difference-based PPO-C, including the impact of $w$ and $\alpha$, as detailed in the Appendix E.6, Tables 10 and 11. A more comprehensive comparison with the threshold-based version is provided in Appendix E.9, Table 15.
> >
> > ---
> >
> > To answer your comment comparing with $w = 0.5$
> >
> > **A1:** Thank you for pointing this out. Our initial reasoning was that when using the difference instead of the absolute value, the adjustment factor becomes smaller, especially during PPO training, where the rewards for policy model responses are generally increasing and predominantly positive. Consequently, we set $w$ to 1.0 initially. To analyze the impact of $w$ on the difference-based PPO-C, we have included the results in Appendix E.6, Table 10. We will also paste below for your reference:
> >
> > | $w$ | MT/Arena-Hard | ECE $\downarrow$ (GSM8K) | AUC $\uparrow$ (GSM8K) | ACC $\uparrow$ (GSM8K) | ECE $\downarrow$ (SciQ) | AUC $\uparrow$ (SciQ) | ACC $\uparrow$ (SciQ) | ECE $\downarrow$ (CommonsenseQA) | AUC $\uparrow$ (CommonsenseQA) | ACC $\uparrow$ (CommonsenseQA) |
> > |---|---|---|---|---|---|---|---|---|---|---|
> > | 0.5 | 8.03 / 14.7 | 0.8792 | 0.521 | 0.1099 | 0.0703 | 0.6031 | 0.896 | 0.1552 | 0.5678 | 0.7674 |
> > | 1.0 | 7.91 / 13.8 | 0.8238 | 0.4937 | 0.119 | 0.0087 | 0.578 | 0.898 | 0.1153 | 0.585 | 0.7625 |
> > | 2.0 | 7.87 / 13.7 | 0.8025 | 0.5342 | 0.1046 | 0.0319 | 0.5892 | 0.906 | 0.0457 | 0.5825 | 0.7699 |
> >
> > ---
> >
> > | $w$ | MT/Arena-Hard | ECE $\downarrow$ (TruthfulQA) | AUC $\uparrow$ (TruthfulQA) | ACC $\uparrow$ (TruthfulQA) | ECE $\downarrow$ (Object Counting) | AUC $\uparrow$ (Object Counting) | ACC $\uparrow$ (Object Counting) | ECE $\downarrow$ (Professional Knowledge) | AUC $\uparrow$ (Professional Knowledge) | ACC $\uparrow$ (Professional Knowledge) |
> > |---|---|---|---|---|---|---|---|---|---|---|
> > | 0.5 | 7.79 | 0.4428 | 0.5549 | 0.4553 | 0.4856 | 0.5036 | 0.512 | 0.4286 | 0.5027 | 0.4906 |
> > | 1.0 | 7.91 / 13.8 | 0.4104 | 0.515 | 0.4492 | 0.4774 | 0.5118 | 0.496 | 0.383 | 0.509 | 0.4902 |
> > | 2.0 | 7.87 / 13.7 | 0.3486 | 0.4856 | 0.4455 | 0.4405 | 0.5309 | 0.509 | 0.3318 | 0.5263 | 0.4798 |
> >
> > ---
> > To summarize: We observe a trade-off between calibration effectiveness and model instruction-following performance. As $w$ increase from 0.5 to 2.0, ECE significantly decreases, while the MT-Bench and Arena-Hard score show slightly reductions. Considering this trade-off and our primary focus on confidence calibration, we select $w = 2.0$ for the main results presented in Table 1.
> >
> > We hope this addresses your concern. Thank you once again for your valuable feedback and suggestions, which have greatly contributed to improving our paper!

---

### Author Response · Authors · 2024-11-19
**Summary of the Update and New Revision of our Paper**

We sincerely thank all reviewers [R1 (Wbvf), R2 (i6YC), R3 (sUqe), R4 (BbuH)] for their thoughtful feedback. We are glad that the reviewers found the problem addressed is relevant and of interest to the community [R1, R3], our motivation is well-supported [R2, R4], our experiments are comprehensive and show clear improvements in calibration metrics [R1, R2, R3, R4], and the paper is well-organized and easy to follow [R1, R2, R3, R4].

Based on the feedback, we have updated our manuscripts accordingly and uploaded a new version of our paper for review. The changes are colored in blue. We summarize the key changes:
* We have fixed typos and revised some sentences as pointed out by reviewers to avoid confusion.
* We have updated and added results for DPO extension using better tuned hyperparameters.
* We have updated Section 4.2 to include a direction comparison of our calibrated reward model and pre-calibrated version.
* We have included more analysis in Appendix:
  * We present the visualization of confidence distribution of our proposed method in Appendix E.4, Figure 22.
  * We present a density plot of model logits for confidence scores at Appendix E.5, Figure 23.
  * We provide experimental results for varying alpha in PPO-C at Appendix E.6 Table 11.
  * We provide experimental results of combining loss in Eq. (1) and Eq. (2) at Appendix E.8, Figure 18 and Table 13.
  * We explore an alternative difference-based approach for PPO-C and provide experimental results at Appendix E.9, Table 14.

---

### Comment · Reviewer_BbUH · 2024-11-30
**Reply to PC**

Dear Program Chairs,

Thank you for the email. However, I would like to point out that the reviewers for submission 5802 have been highly responsive and responsible. Many of us have already acknowledged the author responses and adjusted our scores or confidence as needed, which makes this reminder seem unnecessary.

I kindly suggest focusing such reminders where they are more needed to ensure fairness and efficiency.

Best regards,

Reviewer BbUH

---

### Meta-Review · Area_Chair_5Mhf · 2024-12-20

**Metareview:**

This paper investigates overconfidence in LLMs trained with RLHF, finding that reward models are biased towards high-confidence responses regardless of accuracy. To address this, the authors propose two heuristic methods: PPO-M, which calibrates reward models, and PPO-C, which dynamically adjusts rewards during training. These methods successfully reduce calibration error while maintaining performance comparable to standard PPO across various tasks. The paper's strengths lie in its clear identification of the problem, well-written presentation, and comprehensive evaluation. However, it also has weaknesses, such as limited scientific insight, reliance on verbal confidence, and a lack of theoretical justification. Despite these limitations, I am recommending acceptance based on paper's empirical results and transparent discussion of its limitations.

**Additional Comments On Reviewer Discussion:**

During the rebuttal period, reviewers raised concerns about the paper's scientific contribution, practical value, reliance on verbal confidence, clarity of the confidence metric, and lack of theoretical grounding. The authors responded by providing further analysis and clarification, addressing most of these concerns. They highlighted the insights gained from their work, the advantages of their methods, and the limitations of their approach. While these responses strengthened the paper, the remaining concerns about the heuristic nature of the methods and the lack of strong theoretical support still makes it a borderline submission.

---

### Decision · Program_Chairs · 2025-01-22

Accept (Poster)